# Single-nucleus RNA sequencing reveals glial cell type-specific responses to ischemic stroke in male rodents

Daniel Bormann[1,2], Michael Knoflach[3,4], Emilia Poreba[5], Christian J. Riedl[6,7], Giulia Testa[6,7], Cyrille Orset[8,9], Anthony Levilly[8,9], Andréa Cottereau[8,9], Philipp Jauk [6,7,10], Simon Hametner[6,7], Nadine Stranzl[6,7], Bahar Golabi[5], Dragan Copic [1,2,11], Katharina Klas[1,2], Martin Direder[1,2,12], Hannes Kühtreiber[1,2], Melanie Salek[1,2], Stephanie zur Nedden[13], Gabriele Baier-Bitterlich [13], Stefan Kiechl[3,4], Carmen Haider[6,7], Verena Endmayr[6,7], Romana Höftberger [6,7], Hendrik J. Ankersmit[1,2,14] ✉ & Michael Mildner [5,14] ✉

Neuroglia critically shape the brain´s response to ischemic stroke. However, their phenotypic heterogeneity impedes a holistic understanding of the cellular composition of the early ischemic lesion. Here we present a single cell resolution transcriptomics dataset of the brain´s acute response to infarction. Oligodendrocyte lineage cells and astrocytes range among the most transcriptionally perturbed populations and exhibit infarction- and subtype-specific molecular signatures. Specifically, we find infarction restricted proliferating oligodendrocyte precursor cells (OPCs), mature oligodendrocytes and reactive astrocytes, exhibiting transcriptional commonalities in response to ischemic injury. OPCs and reactive astrocytes are involved in a shared immuno-glial cross talk with stroke-specific myeloid cells. Within the perilesional zone, osteopontin positive myeloid cells accumulate in close proximity to CD44+ proliferating OPCs and reactive astrocytes. In vitro, osteopontin increases the migratory capacity of OPCs. Collectively, our study highlights molecular cross talk events which might govern the cellular composition of acutely infarcted brain tissue.

The brain is among the most metabolically costly mammalian organs[1] and hence particularly vulnerable to ischemia[2]. The sudden deprivation of oxygen and substrate availability in the brain parenchyma triggers a cascade of complex pathophysiological events, culminating in the loss of neural tissue and lasting neurological dysfunction[2,3]. In humans, this oxygen and substrate deprivation is most often caused by an acute, critical reduction of cerebral blood flow, due to the occlusion of large cerebral arteries, the most common cause of ischemic stroke[3].

Ischemic stroke is the second leading cause of disability and death worldwide and the global disease burden of ischemic stroke has been predicted to increase[4]. Apart from supportive care, all currently approved acute treatment strategies, that is thrombolysis and mechanical thrombectomy, aim to reinstate cerebral blood flow and are generally only effective when initiated within a timeframe of under 24 h after stroke onset[5]. Therefore, the lack of treatment strategies directed at neural tissue regeneration constitutes an important unmet therapeutic need. Nevertheless, spontaneous, albeit typically incomplete, regain of function after stroke is common and already observable within the acute phase of recovery, ranging from approximately 1 to 7 days[6,7]. Numerous endogenous recovery mechanisms of the injured CNS have thus been postulated[8].

Cerebral ischemia triggers a breakdown of neurovascular unit (NVU) integrity, inflammation, neuronal cell death, and white matter injury[9,10]. This tissue damage is met with pronounced transcriptional, biochemical, and morphological changes in glial cells, including reactive astrogliosis and early remyelination[10,11]. However, current knowledge on the phenotypic heterogeneity within each reactive cell type and their precise interactions during the acute recovery from cerebral ischemic injury is still limited. Single-cell sequencing technologies have proven to be highly effective in addressing the challenges posed by the complex cellular heterogeneity of the CNS, in health and disease[12]. Arguably, most efforts in dissecting single-cell transcriptomes after cerebral ischemia have been directed at immune and vascular cells[13–18]. Thus far, particularly few studies have captured sufficient oligodendrocyte linage cells to identify robust subtype-specific transcriptional changes following stroke[19,20]. Moreover, extensive transcriptional comparisons between reactivated astrocytes and oligodendrocyte lineage cells in response to cerebral ischemia are still lacking.

Here we present a large-scale single-nucleus transcriptome dataset of the brain´s acute response to ischemic stroke. We dissect subtype-specific transcriptional signatures of stroke reactive neuroglia, compare subtype-specific astrocyte and oligodendrocyte lineage cell responses and contrast these changes with gene expressional profiles found in other CNS injuries. Our study highlights common immuno-glial molecular crosstalk events between myeloid cells, oligodendrocyte precursor cells (OPC), and reactive astrocytes, which might shape the cellular composition and microenvironment during early post-ischemic neural regeneration.

## Results

### MCAO alters CNS cell type composition and induces cell type-specific transcriptional changes

Here we used a rat model of permanent middle cerebral artery occlusion (MCAO) to investigate acute cell type specific transcriptional perturbations, at single-cell resolution in the acute phase following cerebral ischemic injury (Fig. 1a).

The induction of ischemic brain tissue damage was validated by MRI imaging 48 h after injury (Supplementary Fig. 1a−e). Hyper-intense lesions on T2 weighted MRI images were absent from all Sham-operated rats ($n = 4$) (Supplementary Fig. 1b), while animals from the MCAO group ($n = 7$) exhibited pronounced ischemic lesions ranging from 35.01 to 617.2 mm³, which we further stratified into moderate MCAO (mMCAO) ($59.6 \pm 39.2$, $n = 3$) and severe MCAO (sMCAO) ($449.5 \pm 132.5$, $n = 4$) infarctions (Fig. 1a, Supplementary Fig. 1c−e). Importantly, T2 hyper-intense lesions (Supplementary Fig. 2a) closely correlated with myelin and neuronal loss, as evidenced by LFB (Supplementary Fig. 2b) and MAP2 immunofluorescence staining (Supplementary Fig. 2c, d), respectively.

The selection of coronal tissue sections for snRNAseq was guided by MRI data. The maximum extent of the ischemic brain lesions was localized approximately between Bregma anterior-posterior +1.5 mm and −2 mm, in all MCAO samples, thus this region was selected for snRNAseq (Supplementary Fig. 1a−d). Left and right hemispheres were sequenced separately. Hence, we obtained datasets from the left and right hemispheres of Sham-operated rats (Sham L and Sham R, respectively), as well as left (=contralateral to ischemic lesion) and right (=ipsilateral to ischemic lesions) hemispheres of mMCAO and sMCAO infarcted rats (mMCAO contra, mMCAO ipsi, sMCAO contra, sMCAO ipsi, respectively) (Fig. 1a).

Following quality control (Supplementary Fig. 3a−d), we performed unbiased clustering analysis, which grouped all nuclei into 6 non-neuronal and 23 neuronal (12 glutamatergic, 1 cholinergic, 10 GABAergic) major cell clusters (Fig. 1b), using well-established marker genes (Fig. 1c−e). We identified three neuroglia clusters, specifically one immature and one myelinating/mature oligodendrocyte lineage

cluster (OLIGO_1 and OLIGO_2, respectively) and one astrocyte clusters (AC), as well as one ependymal and mural cell cluster (EP_M_C), one vascular cell cluster, enriched for endothelial and pericyte transcripts (VASC) and one myeloid cell cluster (MC). Glutamatergic neurons were broadly split into isocortical, *Satb2* expressing (GLU_Satb2+) and allocortical and deep grey matter, *Satb2* negative glutamatergic neurons (GLU_Satb2−). GLU_Satb2+ neurons were further segregated using cortical layer-specific markers. We identified one cholinergic interneuron cluster (CHOL_IN). GABAergic neurons grouped into various interneuron (GABA_IN) and medium spiny neuron populations (GABA_MSN). GABA_IN were moreover separated into various *Adarb2* positive (GABA_IN_Adarb2+), thus likely caudal ganglionic eminence (CGE) derived and *Adarb2* negative (GABA_IN_Adarb2−), thus likely medial ganglionic eminence (MGE) derived inhibitory interneuron clusters. One GABAergic cluster could not be characterized using known inhibitory interneuron subset specific markers and was thus termed ambiguous GABAergic neuronal cluster (GABA_Amb). A detailed description of this and all following sub-clustering analyses is given in the supplementary notes, nuclei counts and cluster markers are reported in Supplementary Data 1.

Most of the clusters were represented in all datasets (Fig. 1f, Supplementary Fig. 3e). As expected, neuronal clusters were depleted in the dataset derived from severely infarcted tissue (Fig. 1f, Supplementary Fig. 3e). Most notably, almost all captured MC transcriptomes were derived from infarcted tissue (Fig. 1f, g). Their transcriptional signature significantly overlapped with the gene expression profile of stroke-associated myeloid cells (SAMC)[13] (Fig. 1h, Supplementary Fig. 4a) and they expressed both canonical microglia and macrophage, but not lymphocyte markers (Supplementary Fig. 4b). Sub-clustering analyses of the MC cluster revealed two microglia (MG_0, MG_1), three macrophage transcript enriched (MΦe_1 to 3) and one smaller dendritic cell (DC) cluster (Supplementary Fig. 4c, d).

Notably, the expression of SAMC signature genes was well-conserved across MG_1 and MΦe_1 to 3 (Supplementary Fig. 4d, e), suggesting that microglia and macrophages converge onto a common phenotype within infarcted brain parenchyma, as previously reported[13]. We then systematically assessed overlaps between the gene signatures of the stroke-enriched myeloid cell subclusters in our datasets and previously described microglia and macrophage gene expression profiles in normal development and various neuropathologies (Supplementary Data 2). MG_1 and MΦe_1 to 3 exhibited robust enrichment for axon tract-associated microglia (ATM)[21] and disease-associated microglia (DAM) but not disease inflammatory macrophage (DIM)[22] associated transcripts (Supplementary Fig. 4d, e). Furthermore, these clusters overlapped clearly with the transcriptional phenotype of "foamy" microglia enriched in multiple sclerosis (MS) chronic active lesion edges[23], while the profile of iron-associated and activated MS microglia[23,24] was more restricted to MΦe clusters and less prominently represented in our dataset. Likewise, the upregulation of protein synthesis-associated genes (e.g. *Rpl13*, *Rplp1*) typical for iron metabolism associated and activated MS microglia[23,24], was largely restricted to MΦe clusters. Other MS-associated myeloid cell profiles (for example, associated with chronic lesions, antigen presentation and phagocytosis) mapped more diffusely over all myeloid cell clusters (Supplementary Fig. 4e). Lastly, microglia but not macrophage-enriched clusters expressed genes (e.g. *Cdc45*, *Mki67*, *Top2a*) associated to cell proliferation by cell cycle scoring and enrichment analysis (Supplementary Fig. 4d, f). Enrichment analysis of MΦe cluster markers highlighted various degranulation, endo-/phagocytosis, as well as iron and lipid transport and metabolism-related processes and indicated the production of and reaction to reactive oxygen (ROS) and nitrogen species (RNS) (Supplementary Fig. 4g, h). Notably, some of the genes involved in these processes (e.g. *Dab2*, *Lrp1*, *Ctsd*) were also partially enriched in MG_1 (Supplementary Fig. 4i). Interestingly, macrophage transcript enriched clusters were more abundant in severely as compared to

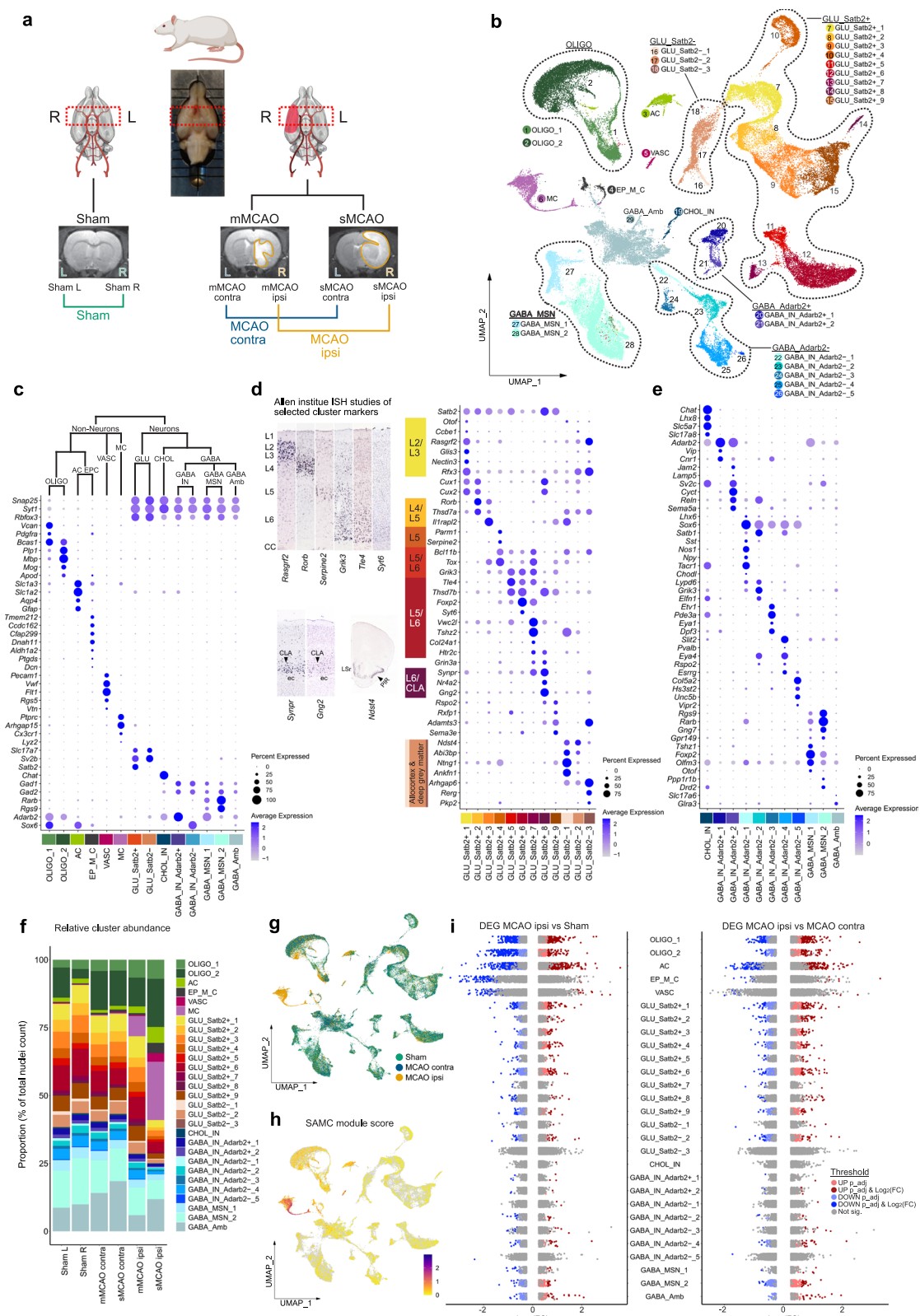

moderately infarcted brain tissue (Supplementary Fig. 5a, b). However, the transcriptional profiles of the individual myeloid cell subclusters did not differ substantially between the two infarction severities (Supplementary Fig. 5c–e). Taken together, these findings underpin the emergence of the SAMC phenotype in the infarcted brain parenchyma and additionally highlight shared and distinct transcriptional signatures of stroke-associated myeloid cell subsets.

We next investigated transcriptional perturbations induced by cerebral ischemia within those major cell clusters, which were represented in all datasets. No major transcriptional differences between the left and right hemisphere derived from Sham control animals were identified (Supplementary Fig. 6a, b). Although, cluster OLIGO_2 contained more nuclei in Sham R, as compared to Sham L (Fig. 1f), 0 DEGs were derived from the comparison of OLIGO_2 transcriptional profiles

**Fig. 1 | snRNAseq reveals differential cell cluster abundance and cluster-specific transcriptional perturbations 48 h after ischemic stroke. a** Illustration of study design, depicting brain regions sampled for snRNAseq, from $n = 4$ Sham control rats and $n = 7$ middle cerebral artery occlusion (MCAO) group rats. MRI images of brain tissue from Sham-operated, moderate (mMCAO) and severe (sMCAO) MCAO group rats are presented, ischemic lesions are highlighted in orange. Rat schematic created with BioRender.com. MCAO ipsi MCAO group, hemisphere ipsilateral to ischemic lesion, MCAO contra MCAO group, hemisphere contralateral to ischemic lesion. **b** Uniform Manifold Approximation and Projection (UMAP) plot depicting 68616 nuclei annotated to 29 major cell clusters in the integrated dataset. Cell cluster AC astrocyte cluster, CHOL_IN cholinergic interneurons, EP_M_C ependymal and mural cell cluster, GABA_Amb Ambiguous GABAergic neuronal cluster, GABA_IN_Adarb2 + , GABA_IN_Adarb2- GABAergic interneurons, *Adarb2* positive/negative, respectively, GABA_MSN GABAergic medium spiny neurons, GLU_-Satb2 + , GLU_Satb2- Glutamatergic neurons, *Satb2* positive/negative, respectively, OLIGO_1 immature oligodendrocyte lineage cluster, OLIGO_2 myelinating and mature oligodendrocyte lineage cluster. **c–e** Dotplots depicting curated marker

genes for all major cell clusters. The dendrogram in (**c**) represents overarching taxons of identified major cell clusters. The dotplot in (**d**) depicts curated cluster markers of glutamatergic neurons. Colored bars next to the gene names denote established associations to cortical layers. Representative corresponding RNA in situ hybridization (ISH) results are depicted next to the colored bars. All RNA ISH studies were taken from Allen Brain Atlas database, and are referenced in detail in Supplementary Tab. 1. L = layer, CLA = claustrum, ec = external capsule, LSr = lateral septal nucleus, PIR = piriform cortex. **e** Dotplot depicting marker gene expression in cholinergic and GABAergic neurons. **f** Stacked bar plot depicting the relative abundance of each cell cluster within each sample. **g** Nuclei distribution coloured by treatment group. **h** Gene module score derived from the stroke-associated myeloid cell (SAMC) gene set[13]. **i** Strip plots depicting distribution of DEGs derived from MCAO ipsi vs Sham and MCAO ipsi vs MCAO contra comparisons, for all major cell clusters. DEGs were calculated using the MAST statistical framework as specified within the Methods section. Source data are provided within Supplementary data.

between both Sham hemispheres (Supplementary Fig. 6a, b). We, thus deemed the differential nuclei abundance in Sham R in this cluster to most likely be a technical artefact. Therefore, the two Sham datasets were pooled in all subsequent analysis. We next separately compared the datasets derived from moderately and severely infarcted hemispheres (mMCAO and sMCAO ispi, respectively) to the pooled Sham dataset. Both comparisons yielded a similar DEG distribution, with astrocyte and oligodendrocyte lineage cells emerging as the most reactive populations (Supplementary Fig. 6c, d). Therefore, and to increase the statistical power and hence robustness of our analysis we pooled the mMCAO and sMCAO datasets and performed subsequent cluster-wise comparisons against the pooled Sham and MCAO contra datasets. The majority of DEGs were derived from neuroglia clusters in both DEG calculations (Fig. 1i, Supplementary Fig. 6e, f). With the exception of OLIGO_1 and OLIGO_2, the gene expression profiles of clusters from the MCAO contra group and their Sham counterparts were mostly similar (Supplementary Fig. 6g). The comparisons of the MCAO ipsi datasets to either Sham or MCAO contra datasets consistently unveiled a higher number of DEGs in excitatory neuronal clusters, as compared to inhibitory neuronal clusters (Fig. 1f, Supplementary Fig. 6.c-f, Supplementary Fig. 7a,b). However, within the MCAO ipsi datasets we noticed the emergence of a canonical cellular stress response signature[25], marked by the upregulation of several heat shock proteins (e.g. *Dnaja1, Hsp90aa1, Hspa8, Hsph1*) in GABA_Amb (Supplementary Fig. 7b). This signature mapped to a discrete subset of this cluster, which upon unsupervised subclustering analysis was revealed to be carried exclusively by misclustered oligodendrocytes but not neurons (Supplementary Fig. 7c–f). Hence, this cluster did not disclose a set of neurons with particular vulnerability to ischemia, but rather underpinned the responsiveness of neuroglia to ischemic injury. A full list of DEGs per cell cluster across all mentioned comparisons is provided in Supplementary Data 3. Collectively, neuroglia ranged among the most transcriptionally perturbed populations within our dataset, 48 h after MCAO.

## Single-nucleus transcriptomics identifies stroke-specific oligodendrocyte lineage cell populations

Neuroglial cells are known drivers of regenerative mechanisms following stroke[10,11], consist of highly heterogeneous subpopulations and are among the most transcriptional perturbed cell populations within our dataset. Therefore, we interrogated these cell populations in more detail. We first jointly sub-clustered OLIGO_1 and OLIGO_2. After manual removal of two clusters with evident neuronal transcript contamination (Supplementary Fig. 8a–d), 10 sub-clusters remained, which could be largely grouped according to canonical developmental stages of the oligodendrocyte lineage trajectory. Specifically, we identified two oligodendrocyte precursor cell clusters (OPC_0, OPC_1),

one committed oligodendrocyte precursor cell cluster (COP), one newly formed oligodendrocyte cluster (NFOLIGO), two myelin-forming oligodendrocyte clusters (MFOLIGO_1 and MFOLIGO_2) and three mature oligodendrocyte clusters (MOLIGO_1 to MOLIGO_3) (Fig. 2a, b). Lastly, one sub-cluster faintly expressed markers of oligodendrocytes and immune cell-associated genes (Fig. 2a, b). Of note, the majority of immune cell transcripts within this sub-cluster were derived from the MCAO ipsi datasets (Supplementary Fig. 8e, Supplementary notes). Importantly, previous research has shown that oligodendrocyte transcripts accumulate in the nuclear compartment of phagocytic myeloid cells, giving rise to clusters expressing both oligodendrocyte and myeloid cell transcripts in vivo[24]. This cluster was thus annotated myeloid cell oligodendrocyte mixed cluster (MC_O-LIGO). Details on marker gene curation are given in the supplementary notes. Notably, the two subclusters, OPC_1 and MOLIGO_1, were predominantly derived from infarcted brain tissue (Fig. 2c, d, Supplementary Fig. 8f).

Pseudotime trajectory analysis, which estimates the dynamic progression of cell states along a differentiation trajectory, indicated that the stroke-specific sub-cluster OPC_1 branched directly from the conserved sub-cluster OPC_0 (Fig. 2e). As expected, the mature oligodendrocyte clusters were associated to the highest pseudo time values. We identified a prominent trajectory bifurcation within MOLIGO_0, with one stroke-specific branch encompassing MOLIGO_1 and one branch extending to MOLIGO_2, which was conserved across all groups. Collectively these analyses suggest the emergence of an infarction-restricted OPC and a mature oligodendrocyte cell state. Notably, cell cycle scoring revealed that sub-cluster OPC_1 was derived from proliferating cells (Fig. 2f).

We next conducted DEG calculations for the oligodendrocyte lineage subclusters which were conserved across all groups (Supplementary Fig. 9a–c). Remarkably, the gene expression profiles of the conserved clusters differed little between the infarcted and contralateral hemisphere, with the exception of MC_OLIGO (Supplementary Fig. 9a), which was enriched in immune process and myeloid cell-associated genes in MCAO ipsi as described above. Likewise, the gene expression profiles of most conserved clusters were similar in the MCAO ipsi and Sham datasets, with the notable exception of OPC_0 (Total DEG: 50), MOLIGO_0 (Total DEG: 185) and MC_OLIGO (Total DEG: 98) (Supplementary Fig. 9b). Interestingly, substantial transcriptional differences between the hemisphere contralateral to infarction and Sham control datasets were only evident in subcluster MOLIGO_0 (Total DEG: 119) (Supplementary Fig. 9c). Importantly, 102 [96,23%] of the downregulated DEGs in the MOLIGO_0 subcluster in MCAO contra relative to Sham, were also identified in the comparison of MCAO ipsi to Sham (Supplementary Fig. 9d) and contained neurexins and neuregulins (e.g. *Nrxn1, Nrxn3, Nrg1, Nrg3*), as well as genes

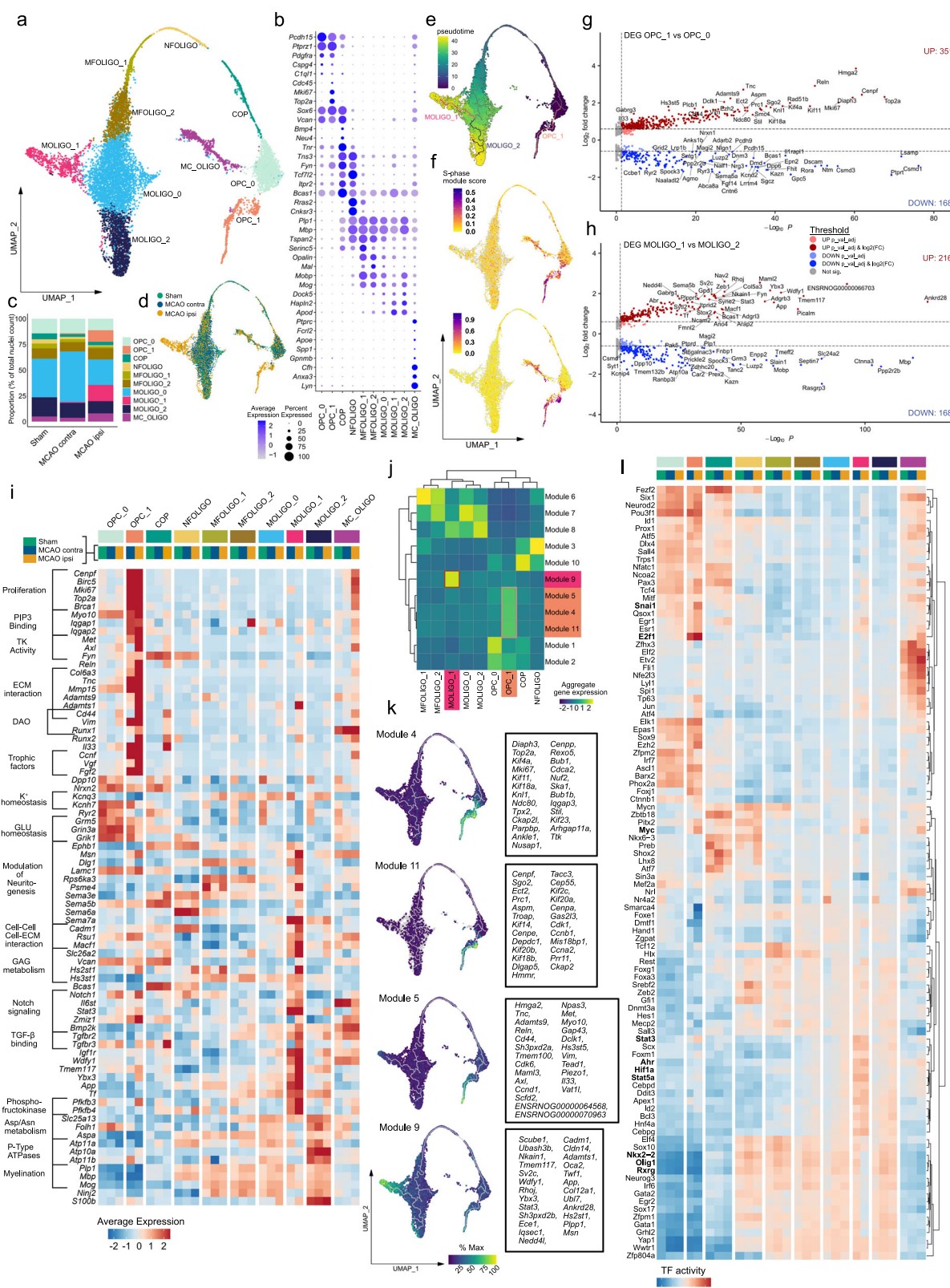

encoding neurotransmitter receptors, ion channels and ion channel interacting proteins (e.g. *Kcnip4*, *Grm5*, *Kcnq5*) (Supplementary Fig. 9e). Of note, the downregulation of many of these genes was subtle in terms of gene expression, as they were a priori expressed at low levels in the Sham dataset within the oligodendrocyte lineage clusters (Supplementary Fig. 9e). All DEGs, derived from all mentioned comparisons are reported in Supplementary Data 4.

We next investigated whether the infarction-restricted oligodendrocyte sub-clusters were conserved across both infarction severities. Importantly, both OPC_1 and MOLIGO_1 were identified in both moderate and severe infarctions, however particularly MOLIGO_1 nuclei were more abundant in severe lesions. All other clusters, as well as the expression of oligodendrocyte lineage cluster markers were well preserved across all datasets (Supplementary Fig. 10a–c).

**Fig. 2 | Emergence of transcriptionally distinct OPCs and mature oligoden-drocytes within infarcted brain tissue. a–d** Subclustering of oligodendrocyte lineage clusters. **a** Uniform Manifold Approximation and Projection (UMAP) plot depicting 10240 nuclei annotated to 10 subclusters. **b** Dotplot depicting curated sub-cluster markers. **c** Stacked bar plot depicting the relative abundance of each subcluster within each group. **d** Nuclei distribution coloured by treatment group. Subcluster OPC oligodendrocyte precursor cell, COP committed oligodendrocyte precursor, NFOLIGO newly formed oligodendrocyte, MFOLIGO myelin-forming oligodendrocyte, MOLIGO mature oligodendrocyte, MC_OLIGO myeloid cell oli-godendrocyte mixed cluster. **e** Projection of Monocle3 generated pseudotime trajectory onto subcluster UMAP plot, with subcluster OPC_0 as root. **f** Feature Plots depicting S-phase and G2/M-phase gene module scores. **g, h** Volcano plots depicting DEGs derived from the comparison of clusters OPC_1 to OPC_0 (**g**) and MOLIGO_1 to MOLIGO_2 (**h**), DEGs were calculated using the MAST statistical fra-mework as specified within the Methods section. **i** Heatmap depicting the average scaled gene expression of curated DEGs, split by subcluster and treatment group.

Functional annotations are given on the left side of the gene names. MCAO ipsi MCAO group, hemisphere ipsilateral to ischemic lesion, MCAO contra MCAO group, hemisphere contralateral to ischemic lesion, PIP3 Phosphatidylinositol (3,4,5)-trisphosphate, TK Tyrosine kinase, ECM Extracellular matrix, DAO Diseases associated oligodendrocyte, GAG Glycosaminoglycan. **j** Clustered heatmap depicting aggregate gene expressions of Monocle3-derived co-regulated gene modules. Modules associated with OPC_1 and MOLIGO_1 are highlighted in orange and magenta, respectively. **k** The average aggregate expression of the OPC_1 and MOLIGO_1 associated modules is plotted along the pseudo-time trajectory. The Top 25 module-defining genes, as sorted by descending Moran´s I, are depicted in boxes on the right side of the respective gene module feature plots. **l** Heatmap depicting the top 100 most variable decoupleR-derived transcription factor activ-ities within the oligodendrocyte lineage sub-clustering analysis, split by sub-cluster and treatment group. Color code on top of the heatmap is defined in (**i**). Source data are provided within Supplementary data.

We next interrogated how the MCAO ipsi-specific sub-clusters (OPC_1 and MOLIGO_1) differed transcriptionally from their homeo-static counterparts, via DEG analyses. Comparisons of OPC_1 to OPC_0 and MOLIGO_1 to MOLIGO_2 within the integrated dataset yielded a total of 519 and 384 DEGs, respectively (Fig. 2g, h, Supplementary Data 4). Conversely, direct cluster-wise comparison between the datasets derived from moderate and severe infarction did not yield substantial transcriptional differences and the key OPC_1 and MOLIGO_1 DEG signatures were well-preserved across both infarction severities (Supplementary Fig. 10d–f). Collectively, these findings indicate that the OPC_1 and MOLIGO_1 cell states emerge in both moderate and severe infarctions.

To gain insight into how the stroke-specific OPC_1 and MOLIGO_1 gene expression signatures might relate to changes in biological function we performed enrichment analyses (summarized in Supple-mentary Figs. 11, 12, Supplementary Data 5). We also systematically compared the signatures of OPC_1 and MOLIGO_1 to each other and to gene expression profiles of diseases associated oligodendrocytes (DAO), derived from various rodent models of neurodegeneration and demyelination[26] (Supplementary Fig. 13, Supplementary Data 2). Transcriptional overlaps between stroke-specific oligodendrocyte lineage sub-clusters with DAO profiles were generally limited (Sup-plementary Fig. 13).

The vast majority of OPC_1 enriched genes mapped to cell cycle progression and proliferation-associated terms (Fig. 2i, Supplementary Figs. 11a–d, 14). Enrichment analyses further highlighted, the upregu-lation of several protein kinases and intracellular scaffold proteins, involved in phosphatidylinositol phosphate (PIP) binding (Supple-mentary Figs. 11a–d), of which many have been associated to OPC cell adhesion, migration, survival and differentiation, such as *Iqgap1*[27], *Met*[28], *Fyn*[29], or *Axl*[30] (Fig. 2i, Supplementary Fig. 14). Several upregu-lated genes, such as *Tnc*, and various metalloproteases indicated extensive interactions of OPC_1 with the extracellular matrix (ECM) (Fig.2i, Supplementary Fig. 14). Notably, the canonical pan-reactive astrocyte markers *Cd44* and *Vim*[31,32], as well as *Runx1* were enriched in both DAO and OPC_1. *Runx2* was likewise upregulated in OPC_1, as well as the neuroprotective immunomodulatory alarmin *Il33*[33]. Interest-ingly, both *Vim* and *Il33*, have previously been shown to be upregulated upon injury in various oligodendrocyte lineage cells[34,35]. Notably, sev-eral growth factors, such as *Ccnf*, *Vgf* and *Fgf2* were also upregulated in OPC_1. Conversely, we observed a downregulation of synaptic transmission-associated transcripts, particularly concerning potas-sium and glutamate homeostasis in OPC_1 (Fig. 2i, Supplementary Fig. 11e–h).

Of note, multiple biological processes associated with MOLIGO_1 enriched DEGs, for example Axonogenesis (GO:0007409), or Axon Guidance (R-HSA-422475) (Supplementary Fig. 12a–c) related to the modulation of neuritogenesis. Several genes encompassed by these

gene sets, such as multiple upregulated sempahorines, have more extensive pleiotropic roles in physiological CNS development and pathology[36]. Similar to OPC_1 several MOLIGO_1 enriched DEGs were associated with ECM, as well as cell-cell interactions and glycosami-noglycan (GAG) metabolism (Fig.2i, Supplementary Fig. 12a–c). Many of the MOLIGO_1 enriched DEGs (e.g. *Dlg1*, *Lamc1*, *Psem4*, *Sema5b*, or *Cadm1*) were also expressed in less mature oligodendrocyte sub-clusters, but absent in the mature oligodendrocyte populations MOLIGO_0 and MOLIGO_2, in the Sham and MCAO contra datasets. Congruently, the expression of the canonical COP and NFOLIGO marker *Bcas1*[37] was markedly higher in MOLIGO_1 as compared to MOLIGO_0 and MOLIGO_2. Thus, several markers of more immature oligodendrocyte developmental stages were uniquely upregulated in MOLIGO_1, but not other mature oligodendrocyte clusters. Several MOLIGO_1 enriched DEGs were related to Notch, TGF-β and IGF-1 sig-nalling, but also included more elusive signalling molecules, like the TIR-domain-containing adaptor (TRIF) recruiter *Wdfy1*[38] (Fig. 2i). Notably, several of the genes associated to Notch signalling by enrichment analyses (Supplementary Fig. 12a, Supplementary data 5), such as *Il6st* and *Stat3* are also crucially involved in multiple type I cytokine signalling pathways[39].

Regarding putative metabolic changes, we noted a robust upre-gulation of the 6-phosphofructo-2-kinase/fructose-2,6-biphosphatase isozyme 3 and 4 coding genes *Pfkb3*, *Pfkb4*, signifying a state of increased anaerobic glycolysis[40]. This was accompanied by a down-regulation of aspartate/asparagine metabolism-related transporters (e.g. *Slc25a13*) and enzymes (e.g. *Folh1, Aspa*) and P-Type ATPases involved in lipid translocation (*Atp10a,11a,11b*) (Supplementary Fig. 12b, d, e, Supplementary data 5). Moreover, several myelination-associated genes were discreetly downregulated in MCAO ipsi-derived MOLIGO_1 transcriptomes as compared to MOLIGO_2 (Fig. 2i). Collec-tively, these findings did not implicate the infarction-restricted MOLIGO_1 cell state in remyelination, but suggest alternative functions.

Complementary to the calculation of DEGs between a priori-defined clusters, we identified genes which changed dynamically along the oligodendrocyte developmental trajectory and combined them into modules of co-regulated genes using Monocle 3. Remarkably, 5 gene-modules mapped uniquely to stroke-specific sub-clusters and mainly consisted of genes which were also identified as OPC_1 or MOLIGO_1 enriched DEGs (Fig. 2j, k, Supplementary data 6). The OPC_1 associated modules 4 and 11 consisted mainly of proliferation-related genes and mapped over the entire OPC_1 cluster. By contrast, module 5, which contained ECM-interaction, migration, survival and immu-nomodulatory process-associated genes appeared further down on the pseudo-temporal trajectory. The aggregate rank score of module 9 increased with incremental distance to the trajectory bifurcation within MOLIGO_0, indicating a dynamic progression from the MOLIGO_0 towards the MOLIGO_1 cell state, specifically within

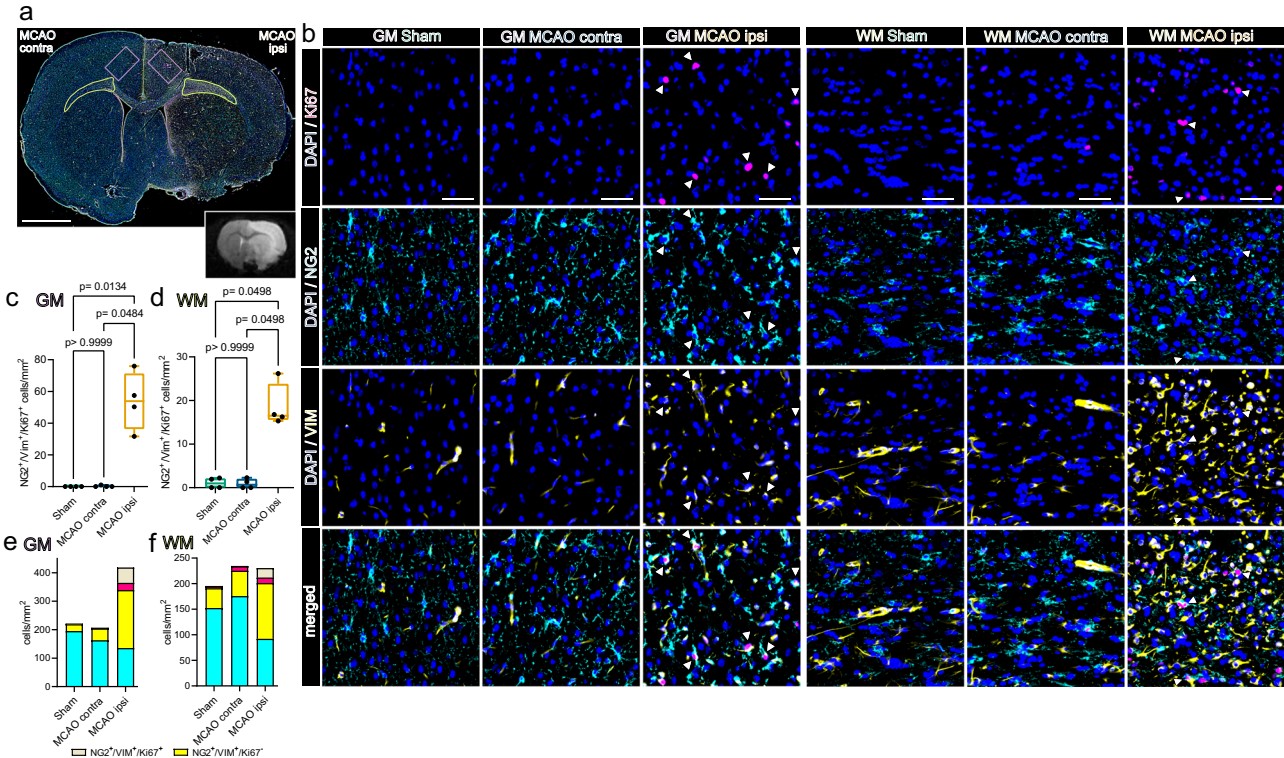

**Fig. 3 | Proliferating, VIM-positive OPCs accumulate in the perilesional zone 48 h after MCAO in rats. a** Overview of a representative coronal rat brain section 48 h post filament induced permanent middle cerebral artery occlusion (MCAO), stained for NG2, VIM and Ki67. Grey matter ROIs (GM) are highlighted in violet, white matter ROIs (WM) in lime green, lower right inset depicts a corresponding T2 weighted MRI image from the same animal. Bar = 2 mm **b** Representative images from GM and WM ROIs of Sham, MCAO contra and MCAO ipsi sections, split by antigen. Ki67 = magenta, NG2 = Cyan, VIM = yellow, DAPI (nuclei) = blue, bars = 50 μm. White arrowheads point to triple-positive cells. MCAO ipsi MCAO group, hemisphere ipsilateral to ischemic lesion, MCAO contra MCAO group, hemisphere contralateral to ischemic lesion. **c–f** Cell counts within GM (**c**) and WM (**d**), respectively, are presented as box plots for NG2$^+$/VIM$^+$/Ki67$^+$ triple positive cells. Box plots depict medians, 25th to 75th percentiles as hinges, minimal and maximal values as whiskers, and individual counts, for each animal as dots. Cell counts for NG2$^+$/VIM$^+$/Ki67$^+$, NG2$^+$/VIM$^-$/Ki67$^+$, NG2$^+$/VIM$^+$/Ki67$^-$, NG2$^+$/VIM$^-$/Ki67$^-$ are also jointly shown as colored stacked bar plots, for GM (**e**) and WM (**f**) ROIs. Data derived from n = 4 animals per group, p values derived from Kruskal–Wallis-H-Tests, followed by Dunn's post-hoc comparisons. Statistical comparison of all subsets within all ROIs is reported in Supplementary Data 7. Source data are provided as a Source Data file.

infarcted tissue. This module was permeated by abundant ECM and cell-cell interaction associated genes (e.g. *Adamts1, Cadm1, Cldn14, Col12a1*), as well as other genes, previously identified as MOLIGO_1 markers during DEG analysis, as described above (e.g. *Stat3, Wdfy1*).

Additionally, we inferred transcription factor (TF) activities using a molecular foot print based approach[41]. In congruence with the previous analyses several TFs for which increased activation was inferred within OPC_1 pertained to proliferation and survival-associated pathways, for example E2f1, or Myc[42] (Fig. 2l). Notable TFs with increased activity in MOLIGO_1 included the hypoxia response related basic helix–loop–helix/Per-ARNT-SIM (bHLH–PAS) superfamily members Ahr and Hif1α[43] and the STAT family members Stat3 and Stat5a. Conversely, the inferred activity of multiple hallmark TFs of oligodendrocyte differentiation and myelination, such as Olig1[44], Nkx2-2[45], or Rxrg[46] was decreased in the MCAO ipsi derived MOLIGO_1 nuclei (Fig. 2l). In summary, using multiple complementary bioinformatics approaches we described the emergence of two transcriptionally unique oligodendrocyte lineage clusters within the infarcted hemisphere, marking the most robust cerebral ischemia induced change within the oligodendrocyte lineage in our dataset.

### Proliferating OPCs accumulate at the perilesional zone

We next conducted Immunofluorescence (IF) staining to confirm the presence of stroke-associated proliferating OPCs in situ and interrogated their spatial distribution in the MCAO infarcted brain, 48 h after permanent filament-based MCAO in rats and thromboembolic MCAO in mice.

Overall, mitosis committed OPCs (NG2$^+$/Ki67$^+$) were almost absent in cortical grey matter (GM) and large white matter (WM) tracts of Sham-operated rats, sparse in the hemisphere contralateral to the infarct lesion, but abundant in the perilesional grey matter and affected white matter (Fig. 3). Furthermore, a substantial number of mitotic OPCs was also positive for VIM (Fig. 3), as predicted by snRNAseq analysis (Fig. 2). Specifically, while essentially absent in the GM and WM of Sham-operated rats (GM: 0/mm$^2$; WM: 1.05 ± 1.2/mm$^2$) and the hemisphere contralateral to infarction (GM: 0.3 ± 0.5/mm$^2$; WM: 0.9 ± 1.1/mm$^2$) the number of NG2$^+$/VIM$^+$/Ki67$^+$ triple positive OPCs increased significantly in the perilesional GM (53.9 ± 18.3/mm$^2$), affected WM (18.6 ± 5.1/mm$^2$) (Fig. 3a–d) and lesion core (11.4 ± 8.2/mm$^2$) (Supplementary Fig. 15a–c). We next investigated whether this OPC phenotype is conserved in an alternative MCAO model, with substantially smaller lesion volumes, in a different species (Supplementary Fig. 16). Congruently, VIM positive, proliferating OPCs, visualized using PDGFRα as an alternative OPC marker (=PDGFRα$^+$/VIM$^+$/Ki67$^+$ triple positive OPCs) accumulated within the perilesional zone 48 h after thromboembolic MCAO in mice, but were only slightly and non-significantly enriched within the lesion core (Supplementary Fig. 17a–h).

Likewise, virtually no NG2$^+$/IL33$^+$/Ki67$^+$ triple positive OPCs were found in the grey and white matter of Sham-operated rats and contralateral hemispheres of MCAO rat brains (Supplementary Fig. 18). In

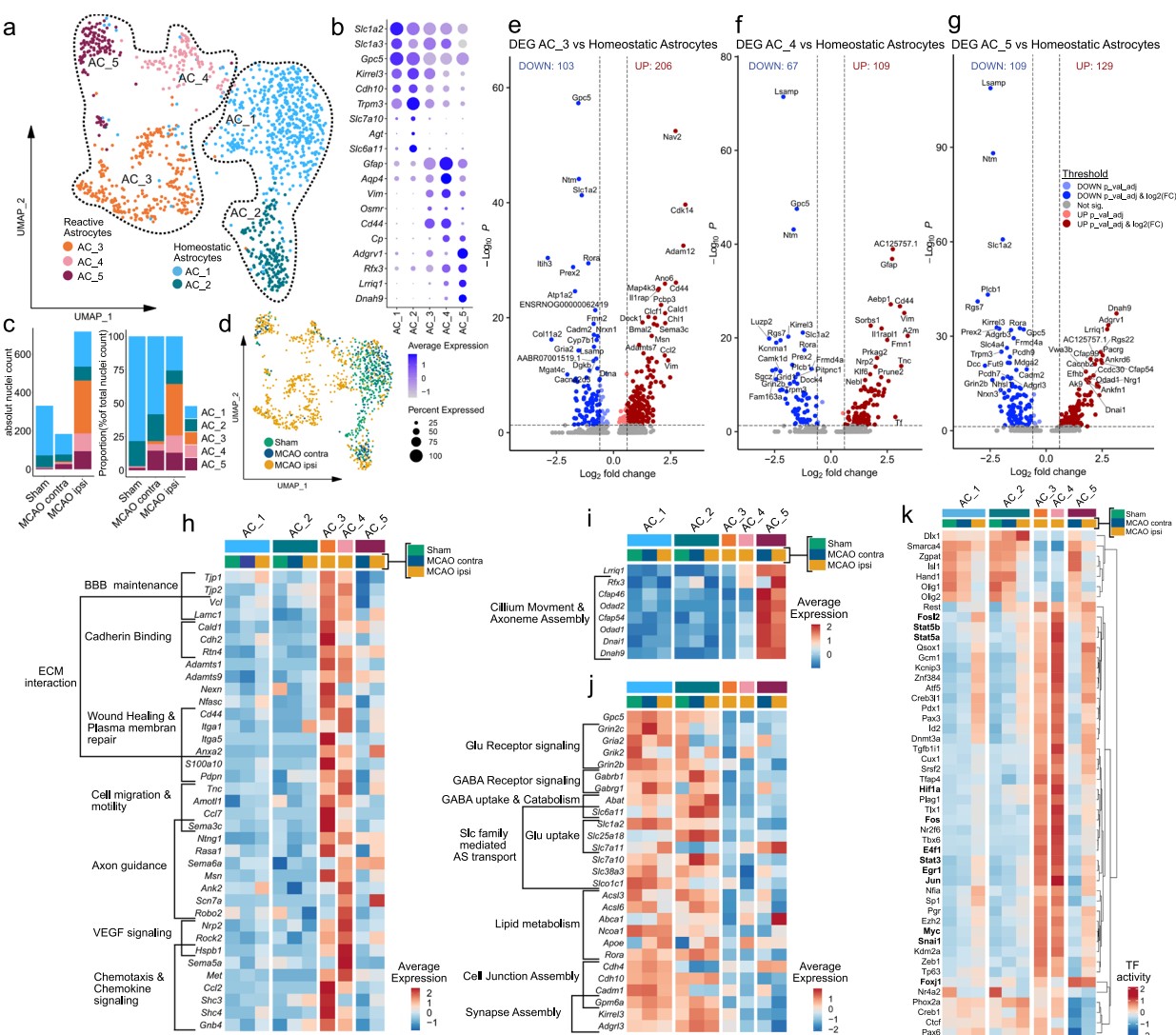

**Fig. 4 | Transcriptional heterogeneity of reactive astrocytes within infarcted brain tissue. a–d** Subclustering analysis of astrocytes. **a** Uniform Manifold Approximation and Projection (UMAP) plot depicting 1233 nuclei annotated to 5 astrocyte (AC) subclusters. **b** Dotplot depicting curated homeostatic and reactive astrocyte marker genes. **c** Stacked bar plots depicting the absolute and relative abundance of each subcluster within each group. MCAO ipsi MCAO group, hemisphere ipsilateral to ischemic lesion, MCAO contra MCAO group, hemisphere contralateral to ischemic lesion. **d** Nuclei distribution coloured by group. **e–g** Volcano plots depicting DEGs derived from the comparison of the reactive astrocyte subclusters AC_3 (**e**), AC_4 (**f**) and AC_5 (**g**) to the homeostatic astrocyte

subclusters (AC_1 and AC_2, pooled). DEGs were calculated using the MAST statistical framework as specified within the Methods section. **h–j** Heatmaps depicting the average scaled gene expression of curated upregulated DEGs, derived from the comparison of AC_3 and AC_4 (**h**) and AC_5 (**i**) to homeostatic astrocytes, as well as DEGs downregulated in reactive astrocytes (**j**), split by subcluster and group. BBB Blood brain barrier, ECM Extracellular matrix, Slc Solute carrier, AS Amino acid (**k**) Clustered heatmap depicting the top 50 most variable transcription factor (TF) activities, within the astrocyte subclustering analysis, split by subcluster and treatment group. Source data are provided within Supplementary data.

contrast, the number of NG2⁺/IL33⁺/Ki67⁺ triple positive OPCs in the perilesional GM and lesion core increased significantly as compared to corresponding Sham GM and regions corresponding to the lesion core (GM: 35.7 ± 26.3/mm² and lesion core: 14.5 ± 10/ mm²) (Supplementary Fig. 18). Similarly, no NG2⁺/IL33⁺/Ki67⁺ triple positive OPCs were identified in the WM of Sham-operated rats, or the WM contralateral to infarction, while they were abundant in affected WM (17.00 ± 5.5/mm²) (Supplementary Fig. 18). However, IL33 was not increased in proliferating OPCs 48 h after thromboembolic MCAO in mice (Supplementary Fig. 19), This suggests that the acute upregulation of IL33 in OPCs is dependent on the species and/or the underlying stroke model and stroke severity. Statistical comparisons of all subsets within all ROIs are reported in Supplementary Data 7, 8.

Collectively, these findings confirmed that mitotic OPCs distinctly accumulate in both the perilesional GM and WM, 48 h after MCAO.

Moreover, a substantial subset of these OPCs expressed the snRNAseq predicted, injury-associated markers VIM, across different species and stroke models, while the upregulation of the injury marker IL33 in OPCs was restricted to filament-based MCAO infarctions in rats.

### Transcriptional heterogeneity of reactive astrocytes in the infarcted brain

Similar to our investigation into the heterogeneous responses to stroke in oligodendrocyte lineage subsets we performed subclustering analysis for astrocytes. After removal of contaminating clusters (Supplementary Fig. 20a–d), we identified 5 astrocyte sub-clusters. 2 subclusters (AC_1 and AC_2) exhibited robust expression of homeostatic astrocyte-associated genes such as *Gpc5, Kirrel3, Cdh10* or *Trpm3*[23,24,47] (Fig. 4a, b). These clusters were identified in all datasets, although their relative abundance decreased slightly in the contralateral hemisphere

of MCAO-operated rats and substantially in the ipsilateral, infarcted hemisphere (Fig. 4c–d, Supplementary Figs. 20e, 21a–c). Conversely, in three subclusters (AC_3 to AC_5) these homeostatic astrocyte markers were expressed more faintly, while they were enriched for pan reactive astrocyte markers such as *Gfap*, *Vim*, *Osmr*, *Cd44*, or *Cp*[31,32] (Fig. 4b, Supplementary Fig. 21b). These three clusters were virtually absent from Sham datasets, sparsely represented in datasets of the contralateral MCAO hemisphere, abundant in infarcted hemispheres (Fig. 4c–d, Supplementary Fig. 21a–c) and thus were annotated as reactive astrocyte clusters. Further details on marker gene curation are given in the supplementary notes. Of note, reactive astrocytes were more abundant in the datasets derived from severe, compared to moderate infarctions (Supplementary Fig. 21a, c). Notably, while the largest reactive astrocyte cluster AC_3 was well-represented in both moderately and severely infarcted tissue, the smaller reactive astrocyte clusters AC_4 and particularly AC_5 were mostly derived from severe infarctions.

To characterize these reactive astrocyte populations in more detail, we compared each reactive astrocyte cluster to the homeostatic subclusters (AC_1 and AC_2) using DEG analyses (Fig. 4e–j, Supplementary Data 9). Notably, the DEG signature of AC_3 was prominent in both moderate and severe infarctions, while the AC_4 and AC_5 reactive astrocyte signature was predominately found in severe infarctions (Supplementary Fig. 21d–i).

Inference of functional characteristics from the DEGs of the reactive astrocyte subclusters (AC_3 to AC_5), using enrichment analyses (summarized in Supplementary Fig. 22–24, Supplementary Data 10) highlighted notable communalities, particularly between the gene signatures of AC_3 and AC_4 (Fig. 4h, Supplementary Fig. 22, 23). For instance, tight (e.g. *Tjp1*, *Tjp2*) and adherence (e.g. *Vcl*) junction components related to blood-brain barrier maintenance were upregulated in both AC_3 and AC_4. Furthermore, several upregulated DEGs in AC_3 and AC_4 related to various ECM interaction, wound healing and cell motility and migration related terms to varying degrees (Supplementary Fig. 22, 23). For example, some cadherin binding related genes (e.g. *Cald1*, *Cdh2*) were more enriched in AC_3, matrix metalloprotease coding genes (e.g. *Adamts1*, *Adamts9*) were upregulated in both, as was *Cd44*. Various reactive astrocyte derived DEGs related to axon guidance and neural cell migration, such as *Sema3c*[48] in AC_3 or *Robo2*[49] in AC_4. Other related to VEGF response (e.g. *Nrp2*, *Rock2*, *Hspb1*) predominantly in AC_4 and chemokine signalling (e.g. *Ccl2*, *Shc3*, *Shc4*) predominately in AC_3.

Overall the transcriptional signature of AC_3 and AC_4 suggest a complex injury response, marked by ECM reorganization, increased migration and involvement in bidirectional communication with other brain parenchymal and infiltrating cell types. AC_5 lacked several of the aforementioned transcriptional features of AC_3 and AC_4, but shared the upregulation of several pan reactive astrocyte markers such as *Gfap*, *Cp*, or *Vim* with the other reactive astrocytes (Fig. 4b, h, i). The most distinguishing characteristic of AC_5 was the enrichment of several gene sets related to cilium and axonemal assembly and movement (Supplementary Fig. 24), including the cilliogenic transcription factor *Rfx3*, cilium dynein arm (e.g.*Dnah9*), or central pair (e.g. *Cfap46*, *Cfap54*) elements[50] (Fig. 4i). Importantly, AC_4 and particularly AC_5 enriched DEGs, were predominantly upregulated in reactive astrocytes in severely infarcted tissue and only to a lesser degree in moderate infarctions (Supplementary Fig. 21g, h). The emergence of these cell states might hence depend on either the overall infarction severity or the lesion of different structural and functional brain areas within the two different stroke severities (Supplementary Fig. 1).

Multiple glutamate (e.g. *Grin2c*, *Gria2*) and GABA (e.g. *Gabrb1*, *Gabrg1*) receptors, glutamate (e.g. *Slc1a2*) and GABA (e.g. *Slc6a11*) reuptake transporters and other solute carrier (SLC) transporters, involved in amino acid import (e.g. *Slc7a10*) were robustly

downregulated in all reactive astrocyte subclusters, within both infarction severities (Fig. 4j, Supplementary Fig. 21i). Several genes related to lipid metabolism (e.g. *Acsl3*, *Acsl6*) and lipid transport (e.g. *Abca1*, *Apoe*) were particularly downregulated in AC_3. Genes involved in synapse assembly and maintenance, such as *Gpm6a* were downregulated in all reactive astrocyte subsets to various degrees. To summarize, reactive astrocytes lost homeostatic gene signatures related to neurotransmitter and lipid metabolism, as well as synapse maintenance in infarcted brain tissue.

The inference of TF activities (Fig. 4k) unveiled further shared patterns in reactive astrocytes. Notable examples of TFs with increased activity in reactive astrocytes related to STAT signalling (e.g. Stat3, Stat5a/b), proliferation, growth and survival (e.g. E4f1, Myc, Jun, Fos, Fosl2), response to hypoxia (e.g. Hif1a) and growth factors (e.g. Egr1)[51], or Snai1, which was recently implicated in the TGF-beta induced glial-mesenchymal transition of Müller glia[52]. Notably, increased activity for multiple of these TFs (e.g. Stat5a, Stat3, Myc, Hif1a, Snai1) was also observed in stroke-specific OPC and Oligodendrocyte subsets (Fig. 2l). Congruent with the upregulation of primary cilium associated genes, increased activity of the cilliogenesis master regulator Foxj1[50] was inferred for AC_5.

Next, we compared the transcriptional signatures of the stroke-associated reactive astrocytes within our dataset to gene expression profiles of reactive astrocytes found in other neurodegenerative and inflammatory neuropathologies (Supplementary Fig. 25). Pan-reactive[31] and neurodegenerative disease-associated astrocyte (DAA)[47] signatures, mapped to several stroke reactive astrocyte subsets (Supplementary Fig. 25a, c–f). No stroke-reactive astrocyte subset in our dataset matched the inflammatory, neurotoxic A1 phenotype, while the neuroprotection-associated A2 signature[31,32] partially overlapped with the signature of AC_3 (Supplementary Fig. 25c, f). Among the reactive astrocyte populations identified in MS (MS_AC_reactive) by Absinta et al.[23], the MS_AC_reactive_1&5 subsets, originally described as "reactive/stressed astrocytes" partially overlapped with the stroke reactive astrocyte clusters AC_3 and AC_4 of our dataset (Supplementary Fig. 25c, d, f). Overlaps with the "astrocytes inflamed in MS" (MS_AIMS) signature were sparse and mainly restricted to pan reactive astrocyte genes (e.g. *Vim*, *Gfap*) (Supplementary Fig. 25c–f). Interestingly, 25 (19.38%) of the 129 DEGs upregulated in AC_5 were also included in the MS_AC_reactive_4 genset, originally described as "senescent astrocytes"[23] (Supplementary Fig. 25e). However, these overlaps did not consist of genes related to senescence, but almost exclusively ciliary process associated genes. Notably, the gene expression profiles of reactive astrocytes and stroke specific oligodendrocyte lineage subsets within our dataset shared extensive similarities (Supplementary Fig. 25g–i). Particularly, we observed that 66 of the 351 DEGs (18.8%) upregulated in OPC_1 and 24 of the 216 DEGs (11.11%) upregulated in MOLIGO_1 were also upregulated in the reactive astrocyte cluster AC_3 (Supplementary Fig. 25g).

## Cell–cell communication (CCC) inference analysis implicates glycoproteins as major immuno-glial signalling hubs in infarcted brain tissue

So far we identified transcriptionally unique myeloid and neuroglial subsets within infarcted brain tissue. Therefore, we next interrogated the molecular cross talk between these cells by inferring potential ligand–receptor (LR) interactions, using LIANA. We only retained the most robust interactions (aggregate rank score ≤0.05) (Supplementary Data 11) and extracted LR pairs unique to datasets from MCAO ipsi. Intriguingly, we inferred 129 LR pairs specific to infarcted brain tissue and grouped them into immuno-glial and intra-glial interactions (Supplementary Figs. 26 and 27, respectively). These interactions corroborated multiple recently inferred stroke response signalling axes, for example between microglia and oligodendrocyte lineage subsets (e.g. *Igf1->Igf1r*, *Thbs1->Cd47*, *Psap->Gpr37*)[19]. Within infarcted

tissue specifically, macrophage-enriched myeloid cell clusters (MΦe) were predicted to signal abundantly via Fibronectin (*Fn1*) onto both myeloid and neuroglia subsets (Supplementary Fig. 26b–d). Cell surface glycoproteins, such as myelin-associated glycoprotein (*Mag*), various integrin and syndecan family members and *Cd44* were the most commonly predicted Fibronectin receptors on myeloid and neuroglial cells (Supplementary Figs. 26, 27). Notably, astrocytes were also predicted to signal via fibronectin -> glycoprotein receptor signalling onto various myeloid and neuroglial subsets (Supplementary Figs. 26e, 27d, e). Glycoprotein receptors indeed emerged as signalling hubs on various myeloid and neuroglial subsets. For instance, *Cd44* was inferred to be targeted by various ECM associated ligands such as fibronectin (*Fn1*), various collagens (e.g. *Col4a1, Col6a3*), *Spp1* encoding osteopontin, but also growth factors, such as hepatocyte growth factor (*Hgf*) or heparin-binding EGF-like growth factor (*Hbegf*) (Supplementary Figs. 26, 27). Microglia and macrophage-derived *Spp1* was predicted to signal back to both myeloid subsets, as well as stroke-specific OPCs (OPC_1) and stroke reactive astrocytes (AC_3 and AC_4) via *Cd44* (Supplementary Fig. 26b–d).

### Cd44 positive reactive astrocytes and proliferating OPCs accumulate at the lesional rim in close proximity to osteopontin-positive myeloid cells

CD44 was identified as a particularly robust marker of reactive astrocytes, in various neuropathological contexts[53] and our dataset (Fig. 4, Supplementary Fig. 25). Surprisingly, we also detected a pronounced upregulation of *Cd44* in stroke associated, proliferating OPCs (Fig. 2). Moreover, *Spp1* -> *Cd44* signalling events from myeloid cells to stroke specific OPCs, reactive astrocytes and myeloid cells themselves ranged among the most robustly predicted interactions within our CCC analysis (Supplementary Figs. 26, 27). Importantly, osteopontin (encoded by *Spp1*)−CD44 signalling was previously implicated in chemotactic cell migration in multiple cell types[54,55]. In situ, a chemotactic attraction of CD44 expressing cells towards osteopontin (OPN) would result in the spatial colocalization of OPN and CD44-positive cells. We thus investigated whether OPN-expressing myeloid cells and CD44 positive neuroglial cells distinctly spatially colocalize in the infarcted brain, using IF stainings.

Indeed, the number of GFAP+/CD44+/VIM+ reactive astrocyte was significantly higher in perilesional cortical GM (123.6 ± 107.9/mm²) as compared to the contralateral GM in MCAO-operated rats (4.2 ± 3.9/mm²) and the GM of Sham-operated rats (2.8 ± 3.3/mm²) (Fig. 5a–c). Likewise, GFAP+/CD44+/VIM- astrocytes were significantly more abundant in perilesional (41.4 ± 21.9/mm²) as compared to contralateral MCAO group GM (4.2 ± 6.1/mm²). Comparison of cell numbers in perilesional to Sham (5.3 ± 3.3/mm²) GM approached significance (Fig. 5c). CD44 was previously implicated as a WM astrocyte subset marker[24]. Congruently, GFAP+/CD44+/VIM- astrocytes were found in all imaged WM ROIs (Sham: 120.3 ± 64.9/mm²; MCAO contra: 116.9 ± 101.8/mm²; MCAO ipsi: 301.4 ± 485.7/mm²) and did not differ significantly between groups. However, the abundance of GFAP+/CD44+/VIM+ astrocytes in the affected WM (626.0 ± 306.8/mm²) increased significantly as compared to MCAO contra (88.4 ± 35.9/mm²) and comparison to Sham WM (120.0 ± 44.04/mm²) approached significance (Fig. 5d). Within the lesion core the number of GFAP+/CD44+/VIM+ astrocytes was not significantly increased relative to anatomically corresponding regions in the contralateral hemisphere, or Sham brain tissue (Supplementary fig. 28a–d). Importantly, these findings were conserved in thromboembolic MCAO in mice (Supplementary Fig. 29a–h).

Proliferating CD44 positive OPCs (NG2+/CD44+/Ki67+) were essentially absent from the GM or WM of Sham-treated rats (GM: 0, WM: 0.25 ± 0.5/mm²) or the hemisphere contralateral to infarction (GM: 0, WM:0), while they were significantly more abundant in perilesional GM (44.3 ± 23.7/mm²), WM (12.3 ± 10.2/mm²) (Fig. 5g–l) as well

as the lesion core (9.7 ± 4.9/mm²) (Supplementary Fig. 28e–h). Congruently, in mice no proliferating CD44 positive OPCs (PDGFRα+/CD44+/Ki67+) were identified within the hemisphere contralateral to infarction, while their number increased significantly within perilesional GM (9.6 ± 2.8/mm²) and WM (25.3 ± 15.2/mm²), but not the lesion core (0.3 ± 0.7/mm²) (Supplementary Fig. 29i–p).

In summary, both reactive astrocytes (GFAP+/CD44+/VIM+) and proliferating CD44 positive OPCs (NG2+/CD44+/Ki67+ or PDGFRα+/CD44+/Ki67+) accumulated in the perilesional zone surrounding the infarcted tissue, across different species and MCAO models.

Within the same region we identified abundant OPN-positive myeloid cells (Iba1+/OPN+), 48 h after MCAO in rats (Fig. 6a–f). Specifically, the number of Iba1+/OPN+ cells within the perilesional GM (161.3 ± 44.4/mm²) was significantly higher as compared to the GM contralateral to infarction (11.6 ± 9.4/mm²) or the GM of Sham-operated rats (17.3 ± 18.4/mm²) (Fig. 6c, e). Likewise, significantly more Iba1+/OPN+ cells were identified in the affected WM (226.3 ± 135.3/mm²) as compared to the WM contralateral to infarction (11.385 ± 18.25/mm²) and the comparison to Sham WM (11.6 ± 9.5/mm²) approached statistical significance (Fig. 6d). Moreover, significantly more Iba1 + /OPN+ cells were identified within the lesion core, compared to the hemisphere contralateral to infarction (Supplementary Fig. 30).

Notably, a substantial amount of OPN expressing myeloid cells was themselves CD44 positive (Fig. 6a–f). In fact, the number of Iba1+/CD44+/OPN+ triple positive myeloid cells was significantly higher in the perilesional GM (142.0 ± 47.5/mm²) as compared to the corresponding GM in the hemisphere contralateral to infarction (1.8 ± 2.4/mm²) or in the GM of Sham-operated rats (2.3 ± 2.6/mm²). Likewise, more Iba1+/CD44+/OPN+ cells were identified in the affected WM (183.05 ± 133.3/mm²) as compared to the WM contralateral to infarction (2.4 ± 3.02/mm²) and the comparison to Sham WM (3.1 ± 2.6/mm²) approached statistical significance. Within infarcted tissue in mice, 48 h after thromboembolic stroke the number of Iba1+/OPN+ cells increased significantly at the perilesional GM and WM boarders, but not within the core lesion as compared to the hemisphere contralateral to infarction (Supplementary Fig. 31). In summary, OPN positive myeloid cells and CD44 positive cells accumulated in close proximity in the perilesional zone, 48 h after infarction, independently of species and MCAO model.

Of note, within infarcted rat tissue we observed that a considerable proportion of CD44 positive myeloid cells was undergoing mitosis within infarcted tissue (Supplementary Fig. 32).

Statistical comparison of all subsets within all ROIs is reported in Supplementary Data 7, 8. Interestingly, the spatial association of OPN positive myeloid cells to bordering CD44 positive cells was also observed in human cerebral infarctions in the stage of advanced macrophage resorption (Supplementary Fig. 33).

### Osteopontin induces OPC migration but not proliferation in vitro

*Spp1* to *Cd44* signaling ranged among the most robustly inferred immunoglial cell−cell interaction events in our dataset (Supplementary Figs. 26, 27). Importantly, increased cellular motility and migration are well-established functional consequences of OPN ->CD44 signaling in multiple cell populations[55,56]. Interestingly, we observed that OPN positive myeloid cells and CD44 positive neuroglia accumulated at the perilesional zone in close proximity in situ (Fig. 6, Supplementary Fig. 31). We thus speculated that OPN might increase the migratory capacity of neuroglial cells. Indeed, OPN was shown to induce migration in astrocytes in vitro[57]. We wondered whether OPN exerts similar effects on OPCs, which was thus far not shown. A cell gap migration assay revealed that the number of OPCs which migrated into the central gap was significantly increased upon OPN treatment (Fig. 7a, b). To exclude that the increased cell number within the central gap 48 h

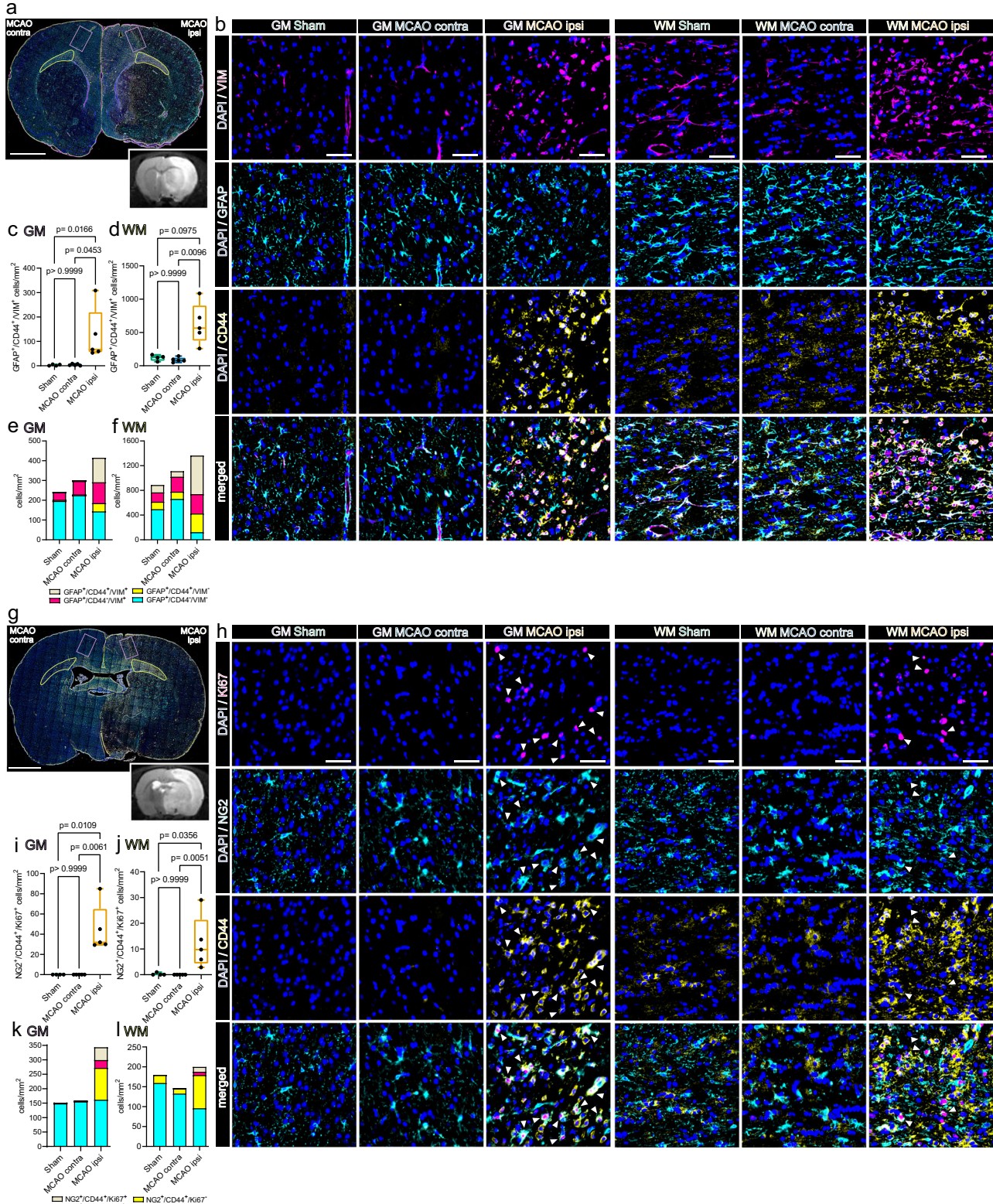

after OPN treatment was caused by enhanced cell proliferation we quantified Ki67 positive cells. The number of Ki67 positive cells within the central gap did not differ significantly between OPN-treated and untreated OPCs (Fig. 7c). We followed up on this observation using a BrdU incorporation assay, as a more sensitive measurement of cell proliferation. Consistently, OPN did not increase the percentage of BrdU positive cells (Fig. 7d). In summary, OPN induces migration but not proliferation of OPCs in vitro.

## Discussion

Reactive astrogliosis is an extensively researched hallmark of the brain´s wound healing response following cerebral ischemia[11,58]. Comparatively, the response of oligodendrocyte lineage cells to ischemic stroke has been less extensively interrogated. Moreover, the phenotypic heterogeneity within each neuroglial subpopulation and their molecular cross talk upon ischemic injury are insufficiently understood, impeding a holistic perspective on the pathobiology of ischemic

**Fig. 5 | Reactive astrocytes and proliferating OPCs are CD44 positive and abundant in the perilesional zone 48 h after permanent MCAO in rats.**
**a** Overview of a representative coronal rat brain section 48 h post middle cerebral artery occlusion MCAO, stained for GFAP, CD44 and VIM. Grey matter ROIs (GM) are highlighted in violet, white matter ROIs (WM) in lime green, lower right inset depicts a corresponding T2 weighted MRI image from the same animal. Bar = 2 mm. **b** Representative images taken from GM and WM ROIs of Sham, MCAO contra and MCAO ipsi sections, split by antigen. VIM = magenta, GFAP = Cyan, CD44 = yellow, all overlaid with DAPI (nuclei) = blue. Bars = 50 µm. MCAO ipsi MCAO group, hemisphere ipsilateral to ischemic lesion, MCAO contra MCAO group, hemisphere contralateral to ischemic lesion. **c–f** Cell counts within GM (**c**) and WM (**d**) are presented as box plots for GFAP+/CD44+/VIM+ triple positive cells. Cell counts for GFAP+/CD44+/VIM+, GFAP+/CD44-/VIM+, GFAP+/CD44+/VIM-, GFAP+/CD44-/VIM- are jointly shown as colored stacked bar plots, for GM (**e**) and WM (**f**) ROIs.
**g** Representative coronal overview, 48 h post MCAO, stained for NG2, CD44, Ki67.

GM ROIs in violet, WM ROIs in lime green, lower right inset shows corresponding MRI image from the same animal. Bar = 2 mm. **h** Representative images from GM and WM ROIs taken from Sham, MCAO contra and MCAO ipsi groups, split by antigen. Ki67 = magenta, NG2 = Cyan, CD44 = yellow. Bars = 50 µm. White arrowheads point to NG2+/CD44+/Ki67+ triple positive cells. **i–l** Cell counts within GM (**i**) and WM (**j**), respectively are presented as box plots for NG2+/CD44+/Ki67+. Cell counts for NG2+/CD44+/Ki67+, NG2+/CD44-/Ki67+, NG2+/CD44+/Ki67-, NG2+/CD44-/Ki67- are also jointly shown as colored stacked bar plots, for GM (**k**) and WM (**l**) ROIs. Data derived from n = 4 Sham control and n = 5 MCAO group animals, p values derived from Kruskal–Wallis-H-Tests, followed by Dunn's post hoc comparisons. All box plots depict medians, 25th to 75th percentiles as hinges, minimal and maximal values as whiskers, and individual counts, for each animal as dots. Statistical comparison of all subsets within all ROIs is reported in Supplementary Data 7. Source data are provided as a Source Data file.

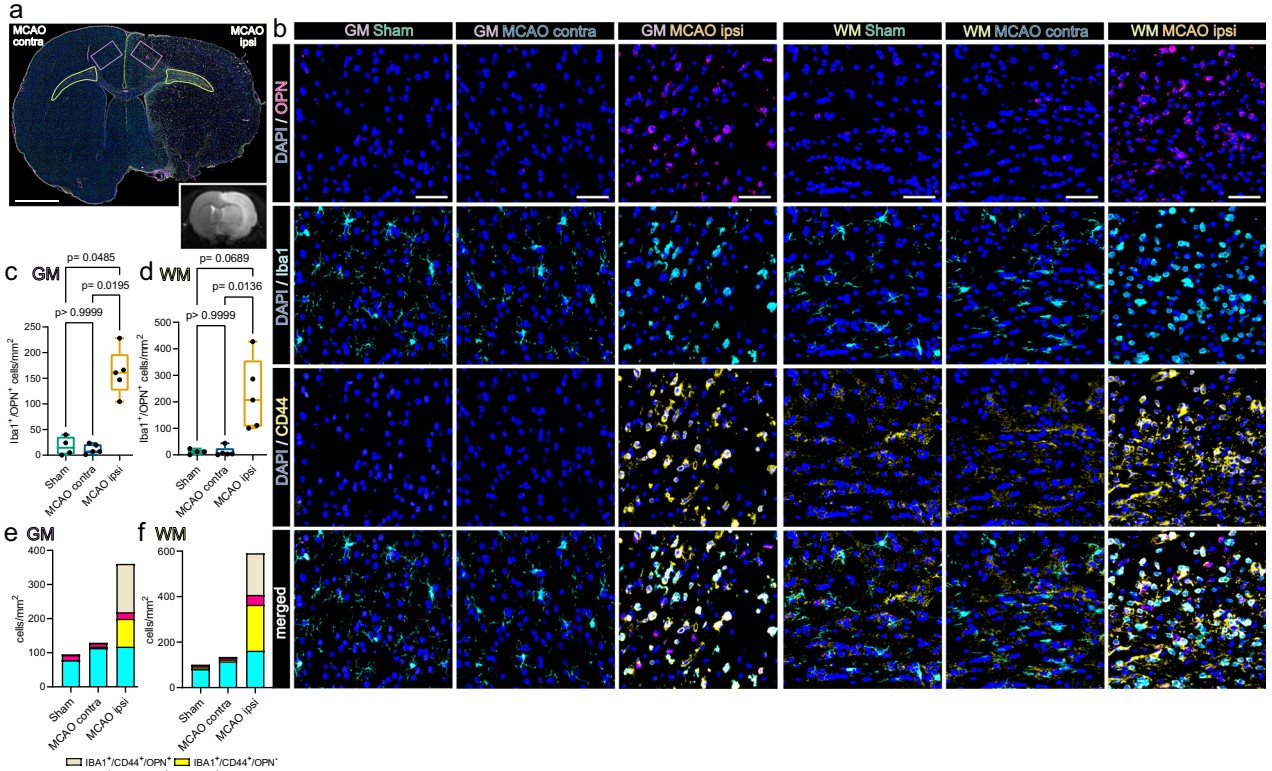

**Fig. 6 | Osteopontin-positive myeloid cells accumulate in the perilesional zone in close proximity to CD44-positive cells 48 h after permanent MCAO in rats.**
**a** Overview of a representative coronal brain section 48 h post MCAO, stained for Iba1, CD44 and osteopontin (OPN). Grey matter ROIs (GM) are highlighted in violet, white matter ROIs (WM) in lime green, lower right inset depicts a corresponding T2 weighted MRI image from the same animal. Bar = 2 mm. **b** Representative images from GM and WM ROIs of Sham, MCAO contra and MCAO ipsi sections, split by antigen. OPN = magenta, Iba1 = cyan, CD44 = yellow. Bars = 50 µm. MCAO ipsi MCAO group, hemisphere ipsilateral to ischemic lesion, MCAO contra MCAO group, hemisphere contralateral to ischemic lesion. **c–f** Cell counts within GM (**c**)

and WM (**d**) are presented as box plots for Iba1+/OPN+ double positive cells, cell counts for Iba1+/CD44+/OPN+, Iba1+/CD44-/OPN+, Iba1+/CD44+/OPN-, Iba1+/CD44-/OPN- are jointly shown as colored stacked bar plots, for GM (**e**) and WM (**f**) ROIs. Data derived from n = 4 Sham control and n = 5 MCAO group animals, p values derived from Kruskal–Wallis-H-Tests, followed by Dunn's post hoc comparisons. All box plots depict medians, 25th to 75th percentiles as hinges, minimal and maximal values as whiskers, and individual counts, for each animal as dots. Statistical comparison of all subsets within all ROIs is reported in Supplementary Data 7. Source data are provided as a Source Data file.

stroke. Here we addressed these challenges by generating a large scale single-cell resolution transcriptomic dataset of the mammalian brain´s acute response to ischemic stroke.

Overall, neuroglial clusters emerged as the most transcriptionally perturbed cell populations within our dataset. Within the oligodendrocyte lineage we detected two transcriptional cell states which were almost uniquely detected within infarcted hemispheres. In line with previous observations[19,59], we found proliferating OPCs at the perilesional zone of the infarcted hemisphere.

However, beyond cell cycle progression the transcriptome of these cells indicated the activation of multiple survival, migration, ECM interaction and growth factor related pathways. This observation suggests pleiotropic roles of OPCs during the brain´s wound healing response to ischemic injury. Indeed, although oligodendrogenesis and hence contribution to myelination are historically the most prominently described features of OPCs, they are increasingly realized to have more multifaceted roles, particularly in response to injury[60].

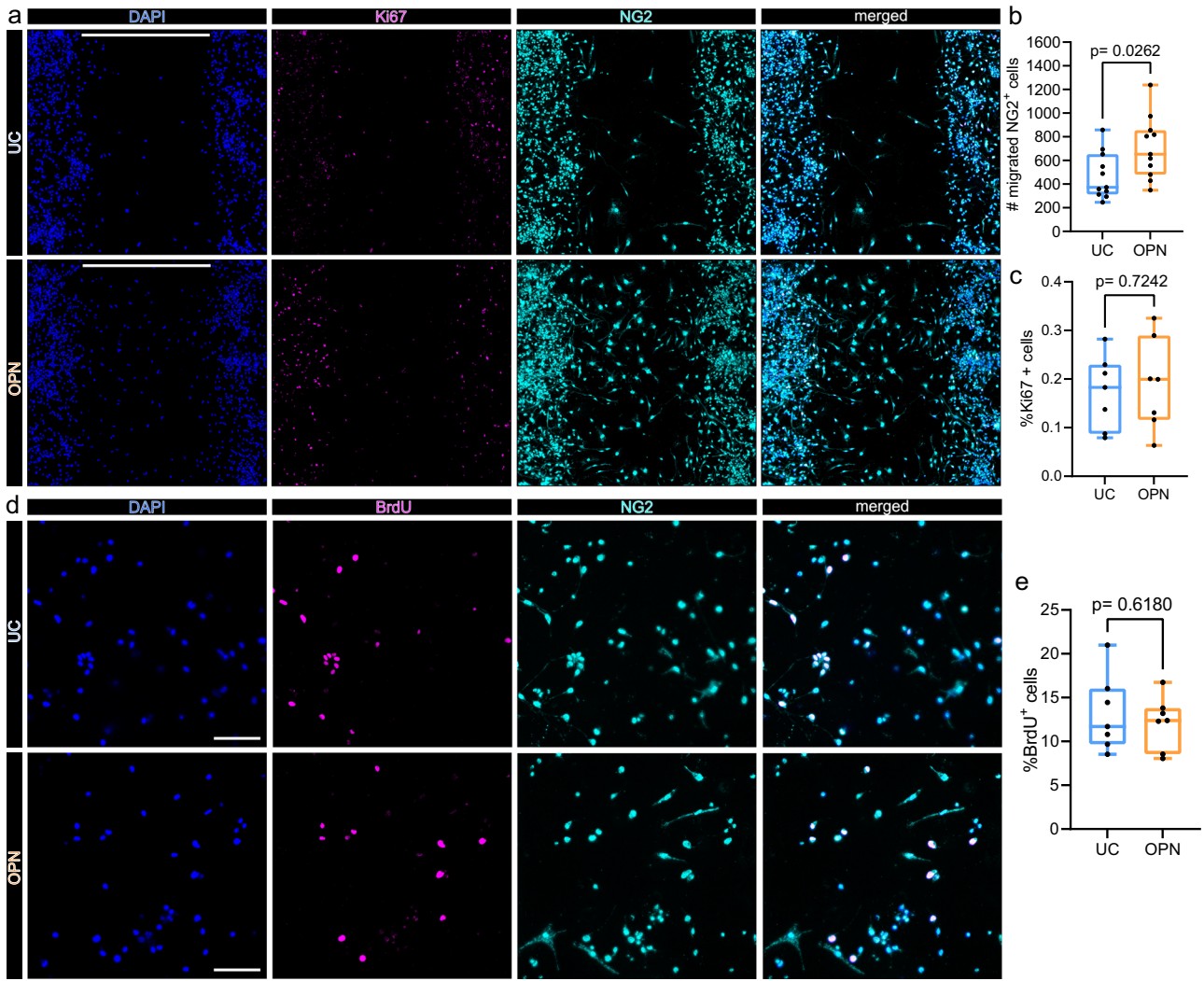

**Fig. 7 | Osteopontin induces OPC migration but not proliferation in vitro. a–c** In vitro cell migration assay. Cells were seeded in 2 well culture inserts, creating defined 500 μm gaps. NG2 positive cells which migrated into the 500 μm gap were quantified after 48 h of treatment. **a** Representative images of OPC cell cultures 48 h after incubation without (upper panel: untreated control = UC), or with 1 μg/ml osteopontin (OPN) (lower panel), stained for DAPI (nuclei) = blue, Ki67 = magenta and NG2 = cyan, split by channel. Scale bars denote 500 μm gaps. **b** Box plot depicting the number of NG2 positive cells, which migrated into 500 μm gaps, for each condition, p-values derived from two-sided unpaired Student's t test ($t = 2.400$, df=20, $n = 11$ replicates per group (independent wells), from 3 independent experiments). In $n = 7$ replicates (independent wells) per group from 2 independent experiments Ki67 was visualized (**c**). **c** Box plot depicting the

percentages of Ki67[+] cells within the 500 μm gap, for each condition, p values derived from unpaired two-sided Student's t test ($t = 0.3612$, df = 12). **d–e** Bromodeoxyuridine (BrdU) incorporation assay. **d** BrdU incorporation was visualized 24 h after incubation without (UC) (upper panel) or with 1 μg/ml OPN (lower panel). Representative x20 magnification images are shown, stained for DAPI (Nuclei) = blue, BrdU = magenta and NG2 = cyan, split by channel. Scale bars = 100 μm. **e** Box plot depicting the percentages of BrdU[+] cells, for each group, p values derived from unpaired two-sided Student's t test ($t = 0.5119$, df = 12, $n = 7$ replicates per group, from 2 independent experiment). All box plots depict medians, 25th to 75th percentiles as hinges, minimal and maximal values as whiskers, and individual counts, for each replicate as dots. Source data are provided as a Source Data file.

The second infarction-associated oligodendrocyte cell state occupied a unique branch at the opposite edge of the oligodendrocyte developmental trajectory. Of note, these oligodendrocytes also upregulated the immature oligodendrocyte marker *Bcas1* and expressed several cell-cell and cell-ECM interaction associated genes, typically enriched in immature oligodendrocytes. This observation might indicate an incomplete ischemic injury-induced reacquisition of immature oligodendrocytes features in a subset of mature oligodendrocytes, a speculation necessitating further investigation. Interestingly, we did not detect a clear myelination-associated gene signature within these cells. On the contrary, several myelin genes (e.g. *Mobp*, *Mbp*) were subtly downregulated and the inferred activity of several myelination and differentiation-associated TFs was decreased in these cells, as compared to homeostatic mature oligodendrocytes. This observation

suggests either that these oligodendrocytes are not yet fully devoted to remyelination at this early post ischemic time point, or that they assume alternative functions during the ischemic wound healing progression. As we have only sampled brain tissue 48 h post-injury, our study is not suited to answer this question conclusively, inviting further investigations, to overcome this limitation. Importantly, recent research has demonstrated a limited and aberrant remyelination capacity of oligodendrocytes surviving demyelination compared to newly formed, progenitor-derived oligodendrocytes[61]. We therefore propose that future work elucidating the potentially divergent fates and remyelination capacities of OPCs and a priori mature oligodendrocytes, following ischemic injury will be crucial.

The transcriptional overlap between the stroke-associated oligodendrocyte lineage clusters within our dataset with previously

described DAO signatures was overall limited, possibly indicating a fine tuned ischemia specific injury response. However, the DAO signatures reported in Pandey et al. (2022) are derived from rodent models of neurodegeneration and de-/remyelination with disease courses ranging from multiple weeks to months[26]. Thus, it is possible that the 48 h post-injury sampling time point in our study was too early to observe the emergence of a more prominent DAO-like signature. Subsequent studies including later sampling time intervals after injury would be advantageous to overcome this limitation of our study. Moreover, other diverging neurodegeneration-associated DAO signatures have been reported and contain further partial overlaps to the stroke-associated oligodendrocyte lineage cells in our dataset[62], for example regarding the upregulation of interleukin 33 (*Il33*). However, within our sample *Il33* was not upregulated in mature stroke associated oligodendrocyte and IL33 upregulation in proliferating OPCs in situ was restricted to filament-induced MCAO infarctions in rats, but not observed after thromboembolic MCAO in mice.

The majority of reactive astrocytes within our dataset upregulated gene sets associated with BBB maintenance, migration, cell-cell and ECM interaction, in line with previous work[63,64]. This transcriptional profile corresponds well to the canonical roles of astrocytes in ECM scaffold formation and spatial containment of the neurotoxic core lesions microenvironment following CNS injury[58]. As expected, the reactive astrocyte signatures within our dataset overlapped with the canonical ischemia-associated A2-signature, but not with the inflammation-induced A1-signature[31,32]. Of note, the stroke reactive astrocyte transcriptomes in our dataset overlapped partially with MS "reactive/stressed astrocytes"[23] and neurodegeneration-associated DAA signatures[47], highlighting common astrocyte responses to diverse neuronal injuries. Notably, we identified a small, predominantly severe infarction-derived, subset of reactive astrocytes (AC_5), characterized by an upregulation of primary cilium-associated genes. Intriguingly, a similar enrichment of primary cilium-associated genes was previously identified in reactive astrocyte subsets in MS and Parkinson's disease[23,65]. Furthermore, Wei et al. have recently characterized a population of astrocyte-ependymal cells in spinal cord tissue which expanded after acute injury[66]. This population shared several transcriptional similarities with the reactive AC_5 subcluster in our dataset such as the upregulation of *Rfx3* and *Dnah* encoding genes and increased Foxj1 TF activity. The precise origin and function of cilia gene enriched or astrocyte-ependymal cell states in response to CNS injuries are still largely unknown. Astrocytes and ependymal cells are developmentally closely related and they form a common transcriptional taxon[67]. This cell state might thus emerge from both a priori astrocytes acquiring ependymal cell features, or ependymal cells upregulating reactive astrocyte-associated genes, or both. Importantly, the AC_5 cell state constituted a smaller reactive astrocyte cluster within our dataset which was largely restricted to severe infarctions. Importantly, our snRNAseq dataset is limited by a singular time point and lack of spatial resolution and thus cannot address these issues unambiguously. Moreover, as astrocytes (n = 1458) only constitute 2,1% of our dataset it is possible that our study does not capture the full in vivo transcriptional heterogeneity of reactive astrocytes in infarcted tissue. Further studies will thus be necessary to unveil the elusive role of these cells in ischemic stroke and other neuropathologies.

Under physiological conditions astrocytes are crucially involved in the regulation of ion and neurotransmitter signalling, as well as synapse assembly and maintenance[68]. Our findings indicate a possible loss of these homeostatic functions in stroke-reactive astrocytes. Likewise, oligodendrocyte lineage cells are increasingly recognized to be coupled to neuronal neurotransmitter signalling via bidirectional cross talk, at OPC and axo-myelinic synapses[60,69]. Notably, similarly to astrocytes, multiple neurotransmitter receptors, ion channels and ion channel interacting proteins were downregulated in oligodendrocyte

lineage cells upon cerebral ischemia. Some of these genes were also downregulated subtly in the hemisphere contralateral to infarction. Further studies will be necessary to assess whether these changes functionally relate to possible disruptions of homeostatic oligodendroglia-neuronal crosstalk.

While comparing the cerebral ischemia-induced transcriptional perturbations in oligodendrocyte lineage cells and astrocytes we noticed further, more prominent similarities between infarction-restricted oligodendrocyte lineage and astrocyte populations. Similar to reactive astrocytes, infarction-restricted OPCs upregulated migration, cell-cell and ECM-interaction genes (e.g. *Met*, *Cdh2*, *Tnc*, *Adamts1*, *Adamts9*, *Vim*, *Cd44*) and colocalized with reactive astrocytes at the perilesional zone. It is thus possible that the perilesional microenvironment instructs a shared phenotype onto these populations. The partial acquisition of reactive astrocyte-associated genes in oligodendrocytes after injury has also been noted in previous studies. Importantly, Kirdajova et al. have documented the emergence of a stroke specific, transient, proliferating "Astrocyte-like NG2 glia" population seven days after focal cerebral ischemia and speculated that this population might contribute to early glial scar formation[35]. Although, a subset of OPCs progresses to myelinating oligodendrocytes following ischemic stroke particularly in young animals[19], a substantial amount of OPCs remains undifferentiated for up to eight weeks post-injury, suggesting potential alternative cell fates[59].

More recently it was also shown that bona fide mature oligodendrocytes can dedifferentiate via a hybrid "AO cell" state into astrocytes in vivo, in the days to weeks following traumatic and ischemic brain injury[70]. This phenotypic switch was causally linked to IL6 signalling. Interestingly, the infarction enriched mature oligodendrocytes in our dataset would be primed to respond to this cytokine due to the prominent upregulation of canonical downstream targets of IL6, such as *Il6st* and *Stat3*. However, the transcriptional similarities between reactive OPCs, oligodendrocytes and astrocyte cluster within our datasets do not necessarily indicate that they harbour the progeny of hybrid cell states. Moreover, overlaps in gene and TF signatures do not unequivocally dictate shared functions during neural regeneration. For example, traumatic injury induced STAT3 activation in astrocytes is involved in GFAP upregulation, induction of cellular hypertrophy and glial scar formation[71], while in oligodendrocytes STAT3 signalling was implicated in maturation and remyelination after focal demyelination[72]. Importantly, our study only captures the transcriptome at 48 h after MCAO. Due to this limitation the herein presented data cannot resolve the ultimate fate of the herein described reactive neuroglial cell states and their ultimate role in subsequent stages of subacute and chronic post-stroke recovery. Therefore, further studies will be necessary to decipher the precise functional consequences of the herein described gene expressional changes in oligodendroglia and astrocytes.

The recruitment of reactive neuroglia and immune cells to the injury site is a crucial step during the acute ischemic injury response[73]. Our data indicate that myeloid cells might be involved in orchestrating this process. We confirmed the emergence of SAMC specifically within infarcted brain tissue. The sparsity of myeloid cells within non-lesioned brain tissue is a previously elaborated[74] limitation of our nuclei isolation approach and impeded a direct comparison of SAMC to homeostatic microglia within this study, which has already been conducted elsewhere[13]. Although the herein used method of nuclei isolation is prone to exclude immune cells, they were robustly captured within infarcted tissue, highlighting their drastically increased abundance in the infarct lesion. The clearance of lipid-rich tissue debris has been established as a primary function of SAMC[13], although alternative functions, such as the reduction of ROS stress, have been described[14]. Here we observed that myeloid cells expressing the canonical SAMC marker osteopontin accumulate in close proximity to reactive astrocytes and proliferating OPCs which robustly expressed the

osteopontin receptor CD44 in the perilesional zone. Furthermore, we were able to show that osteopontin increased the migratory capacity of OPCs in vitro. Indeed, osteopontin is a well-established inductor of cellular migration in numerous cell types and has been implicated to act as a chemotactic cue via CD44 mediated signalling[56,57]. In addition, CD44 was shown to be indispensable for the migration of transplanted rat OPC-like CG4 cells towards focal demyelinated lesions[75], and macrophage-derived osteopontin was shown to induce the extension of astrocyte processes towards the infarct perilesional zone following focal cerebral ischemia[76]. The accumulation of CD44 positive astrocytes and immune cells in the peri-infarct zone was previously shown and associated to a homing towards the CD44 ligand hyaluronic acid, which also accumulates at the peri-infarct border[77–79]. Importantly, our CCC analysis inferred that CD44 is indeed targeted by multiple ligands in infarcted tissue specifically. Likewise, osteopontin was predicted to signal onto multiple other receptors, such as integrins which was previously also associated with increased migration[80]. As the recruitment of neuroglia and immune cells to the site of injury is a prerequisite for further regenerative mechanisms, it is highly plausible that multiple redundant mechanisms have evolved to achieve this. Moreover, the role of osteopontin in ischemic stroke likely exceeds the regulation of cellular migration, although its precise contribution to post-ischemic regeneration is still controversial. For example, osteopontin has been shown to acutely aggravate ischemia-induced BBB disruption[81], but augment white matter integrity via immunomodulatory mechanisms, in the subacute to chronic stages of stroke recovery[15]. A further complicating factor is the age dependency of many molecular cross talk events during the response to cerebral ischemia. For example, osteopontin to CD44 signaling was largely restricted to young animals in a previous study[19]. The fact that our results are based on a homogenous cohort of young, male rodents is an evident limitation that has to be taken into account when extrapolating translational considerations from this dataset.

In summary, our study captured the emergence of cell type and cerebral infarction-specific transcriptional signatures in neuroglia. Although, reactive oligodendrocyte lineage cells and astrocytes exhibited distinct responses their transcriptional signatures overlapped substantially, indicating a shared molecular ischemia response repertoire and possibly shared functions during regeneration. Moreover, we uncovered a shared immuno-glial molecular cross talk, which implicated myeloid cells as contributors to OPC and reactive astrocyte recruitment to the injury site via the osteopontin CD44 signaling axis. Beyond the diverse transcriptional response patterns highlighted in our analysis, the large-scale dataset generated within this study will provide an instrumental resource for the interrogation of acute cell type-specific responses to ischemic stroke. We propose, that this approach will contribute to untangle the complex mechanisms governing post-ischemic neural regeneration, ultimately aiding in the discovery of novel treatment strategies to alleviate the devastating consequences of ischemic stroke.

## Methods

### Study approval/ethics statement

All in vivo animal experiments were performed in accordance with the French ethical law (Decree 2013-118) and the European Union directive (2010/63/EU). The protocol was submitted for ethic approval to the French Ministry of Research and the ethical committee (CENOMEXA—registered under the reference CENOMEXA-C2EA−54) and received the agreement numbers #36435 and #36135. The experiments have been reported in compliance with ARRIVE 2.0 guidelines. Archived human biopsy-derived brain tissue, which was provided by donors upon informed consent, was used in agreement with the Medical University of Vienna ethics committee votes: EK1636/2019, EK1454/2018. Donors did not receive financial compensation for consent.

### Animal husbandry

Experiments were performed on male Wistar rats (6 weeks at receipt, ± 30 g, Janvier Lab, Le Genest-Sainte-Isle) and C57BL/6J mice (6–7 weeks at receipt, BW: 20–25 g, Janvier Lab, Le Genest-Sainte-Isle). All animals included in in vivo experiments were male, to reduce a priori biological variability within the study cohort. Throughout the experiments, animals were maintained in standard husbandry conditions (temperature: $22 \pm 2\,°C$; hygrometry: $50 \pm 20\%$), under reversed light-dark cycle (light from 08:00 to 20:00), with ad libitum access to water and food. Rats were housed at two per cage and mice at five per cage in the presence of enrichment.

### Permanent middle cerebral artery occlusion (MCAO) model in rats

Cerebral ischemia was induced by intraluminal occlusion of the middle cerebral artery (MCAO). Briefly, rats were anesthetized with isoflurane (2-2.5%) in a mixture of $O_2/N_2O$ (30%/70%). During surgery, animal temperature was monitored with a rectal probe and was maintained at $37.5\,°C$ with a heating pad. To induce permanent occlusion of the middle cerebral artery (MCA) a silicone rubber-coated monofilament (size 5-0, diameter 0.15 mm, length 30 mm; diameter with coating 0.38 +/− 0.02 mm; Doccol, Sharon, MA, USA) was introduced into the lumen of the right external carotid, advanced through the internal carotid, and gently pushed up to the origin of the MCA. After wound stitching, the rats were returned to their home cage after receiving analgesics (buprenorphine, 0.05 mg/kg, subcutaneously). In Sham-operated animals all experimental procedures were performed except for the filament insertion.

### Thromboembolic stroke model in mice

Permanent thromboembolic middle cerebral artery (MCA) occlusion was performed as originally described[82]. Briefly, under aseptic conditions, mice were anesthetized with isoflurane (5%) and maintained with 2% isoflurane in a 70%/30% mixture of $NO_2/O_2$. Buprenorphine was injected subcutaneously (0.5 mg/kg) and local anesthesia was performed by instillation of lidocaine in both ears. During the surgical procedure, animal temperature was monitored using a rectal probe and a heating pad was used to maintain body temperature at $37\,°C$. Mice were fixed in a stereotaxic device and the right temporal lobe muscle was retracted, followed by craniotomy. After excision of the dura the right MCA was exposed. A glass micropipette filled with 1 μl murine alpha thrombin (1 UI, Stago, NL) was introduced into the MCA lumen. Then, in situ clot formation was induced by thrombin injection. After 10 min, the clot was stabilized and the pipette was removed. Buprenorphine (0.5 mg/kg) was subcutaneously administered to ensure analgesic coverage before waking up.

### Magnetic resonance imaging (MRI)

To confirm successful induction of ischemic stroke and determine the anatomical localization of the stroke lesion MRI was carried out 48 h after stroke onset, on a Pharmascan 7 T MRI system, using surface coils (Bruker, Germany), following a previously described approach[83]. For lesion volume evaluation, T2-weighted images were acquired using a multislice multiecho sequence: TE/TR 33 ms/2500 ms. Lesion sizes were quantified on these images using ImageJ software. Lesion volumes were determined by a trained investigator blinded to condition and are expressed in $mm^3$.

### Tissue sampling

After completion of MRI studies, animals were sacrificed via sharp blade decapitation in isoflurane anaesthesia, as described above. For single nucleus RNA sequencing studies whole rat brains were extracted and swiftly cut into standardized coronal sections using an adult rat brain slicer matrix (BSRAS003-1, with 3 mm coronal section intervals, Zivic Instruments, Pittsburgh, PA, USA) and hemispheres were

separated. Coronal slices, separated by hemisphere were then immediately snap frozen in liquid nitrogen and stored at −80 °C until further transport on dry ice.

For Immunofluorescence (IF) assays anesthetized animals were transcardially perfused with DPBS, followed by perfusion with 4%PFA in DPBS, brains were harvested whole, further post fixed overnight in 4%PFA in DPBS, and washed three times in DPBS. Brains were then stored in DPBS with 0.05% Sodium Azide at 4 °C until further processing. To match the anatomical regions used for snRNAseq assays, rat brains where cut into standardized coronal sections using the same adult rat brain slicer matrix (BSRAS003-1), described above. From murine brains coronal section between Bregma anterior-posterior −0.8 and +0.8 were used for IF assays. After cutting, brain tissue was dehydrated and embedded in paraffin.

## Single-nuclei preparation

Single-nuclei suspensions were prepared as previously described[74]. Briefly, frozen brain sections were thawed in ice cold Nuclei Extraction Buffer (cat#: 130-128-024, Miltenyi) in gentleMACS™ C-Tubes (cat#: 130-093-237, Miltenyi Biotec, Bergisch Gladbach, Germany), followed by automated gentleMACS™ Octo dissociation (cat#: 130-096-427, Miltenyi) using program: 4C_nuclei_1 and a further 6 min incubation on ice. Suspensions were then strained into 15 ml polypropylene tubes (cat#: 430766, Corning, Corning, NY, USA) over 70 μm strainers (cat#: 542070, Greiner Bio-One International GmbH, Kremsmünster, Austria), 4 ml of ice cold nuclei extraction buffer were added, followed by centrifugation at 500 g, 4 °C, for 5 min on a swing bucket centrifuge (Allegra X-12R, Beckman Coulter, Brea, CA, USA). The supernatant was decanted and the pellet was resuspended in 0.25% (vol/vol) Glycerol (cat#: G5516, Sigma Aldrich) and 5% (wt/vol) bovine serum albumin (BSA) (cat#A-9647, Sigma Aldrich) in Dulbecco's phosphate-buffered saline (DPBS) (cat# 14190-94, Gibco, ThermoFisher Scientific, Waltham, MA, USA)) ( = nucleus wash buffer (NWB1), buffer composition derived from[84]. Suspensions were then strained through 40 μm strainers (cat#: 352340, Falcon®, Corning) and centrifuged at a swing bucket centrifuge at 500 g, 4 °C for 5 min. The pellet containing nuclei and debris was resuspended in a Tricin-KOH buffered (pH 7.8), 10% Iodixanol solution (10% Iodixanol (OptiPrep™, cat#: 7820, STEMCELL Technologies, Vancouver, BC, Canada), 25 mM KCl (cat#: 60142), 5 mM MgCl2 (cat#: M1028), 20 mM Tricin (cat#: T0377) KOH (cat#: 484016), 200 mM Sucrose (cat#: S0389), all from Sigma Aldrich) and gently layered on top of a 20% Iodixanol gradient cushion (20% Iodixanol, 150 mM Sucrose, 25 mM KCl, 5 mM MgCl2, 20 mM Tricine-KOH, pH 7.8) in 14×89 mm thin wall polypropylene centrifuge tubes (cat#: 344059, Beckman Coulter, Brea, CA, USA). An Optima L-80 Ultracentrifuge (serial#: Col94H18, Beckman Coulter), with swing bucket SW41 Ti cartridges, precooled to 4 °C was used for gradient centrifugation at 10000 g, for 30 min, with maximal acceleration and no brake. Following centrifugation, debris fractions were discarded and the purified nuclei pellet was resuspended in ice cold NWB1 and strained over 30 μm strainers (cat#: 130-098-458, Miltenyi). The suspension was centrifuged at a swing bucket centrifuge at 500 g, 4 °C, for 5 min, supernatant was discarded and nuclei were resuspended in a solution of 3% BSA, 0.125% Glycerol, in DPBS ( = NWB2). This washing step was repeated once. Finally, nuclei were resuspended in a solution of 1.5% BSA in DPBS on ice. To obtain nuclei counts, nuclei were stained using the Acridine Orange/Propidium Iodide (AO/PI) Cell Viability Kit (cat#: F23001, Logos Biosystems, Anyang-si, Gyeonggi-do, South Korea). Nuclei were counted as PI positive events using a LUNA-FL™ Dual Fluorescence Cell Counter (cat#: L20001 Logos Biosystems). The fraction of non-lysed Acridine Orange + cells was <5% in all samples. All buffers used during nuclei purification were supplemented with 0.2 U/μl RiboLock RNase Inhibitor (cas#: EO0384, ThermoFisher Scientific, Waltham, MA, USA).

## Single-nucleus processing and library preparation

Processing of single-nuclei suspensions was performed as previously described[74], using the Chromium™ Next GEM Single Cell 5' Kit v2 (PN-1000263, 10 × Genomics, Pleasanton, CA, USA), as per manufacturer´s protocols (CG000331 Rev D, 10 × Genomics). In brief, for Gel Beads-in-Emulsion (GEMs) generation we loaded nuclei onto Chromium™ Next GEM Chips K (PN- 1000286,10 × Genomics), aiming at a recovery of 10-12 × 10³ nuclei per lane, followed by GEM reverse transcription (GEM-RT) and clean up. GEM-RT products were subjected to 14 cycle of cDNA amplification using 10X poly(dT) primers, followed by 10 × 5' gene expression library construction. The Single Index Kit TT Set A (PN 1000215, 10X Genomics) was used for sample indexing during library construction. SPRIselect Reagent Kit (cat#: B23318, Beckman Coulter) beads were used for clean-up procedures, as per 10X protocols instructions. The quality of the obtained libraries was assessed using a DNA screen tape D5000 on a TapeStation 4150 (Agilent Technologies, Santa Clara, CA, USA) and cDNA was quantified using a Qubit 1xdsDNA HS assay kit (cat#: Q33231) on a QuBit 4.0 fluorometer (Invitrogen, ThermoFisher Scientific). Libraries with unique indices were then pooled in equimolar ratios before sequencing.

## Sequencing, pre-processing, and quality control

Samples were sequenced paired-end, with dual indexing (read length 50 bp) using a NovaSeq 6000 (Illumina, San Diego, CA, USA). All samples were processed on the same flow cell. Raw gene counts were obtained by demultiplexing and alignment of reads to the most current *rattus norvegicus* reference genome mRatBN7.2, using the Cellranger v.7.0.0 pipeline, including intronic reads in the count matrix to account for unspliced nuclear transcripts, as per developer's recommendations. Cellranger outputs were further processed utilizing R and R Studio (R version 4.2.2, The R Foundation, Vienna, Austria), using the below indicated packages. Unless otherwise stated, all computational snRNAseq analyses were carried out within the environment of the Seurat package v.4.3.0[85], as per the developer's vignettes.

For each individual dataset UMI count matrices were generated and converted to Seurat Objects, preliminary normalization and variance stabilization was performed using the SCTransform, v2 regularization[86,87], followed by PCA dimensionality reduction with 50 principal components, and graph-based clustering using the "RunUMAP" "FindNeighbors" and "FindClusters" commands.

Using the preliminary clustering information for each dataset, ambient RNA contamination was estimated and ambient RNA was removed using the SoupX v1.6.2 package[88], following developers vignettes. The decontaminated expression matrices were then further processed following the standard Seurat quality control pipeline. Briefly, nuclei with <500 UMI counts, <250 or >5000 expressed genes and >5% mitochondrial genes expressed, were removed from downstream analysis. Doublets were estimated and removed using the DoubletFinder v2.0.3 package[89], as per developers vignettes. All genes with less than 3 UMI counts per feature and all mitochondrial genes were removed from downstream analyses.

## Dataset integration

After the above-described quality control pipeline, normalization and variance stabilization was performed for all individual datasets, utilizing SCTransform, with v2 regularization, with the percentage of mitochondrial reads "percent.mt" passed to the "vars.to.regress" argument. All datasets were then integrated using reciprocal PCA (RPCA) based integration. Briefly, the top 3000 highly variable genes were selected utilizing the "SelectIntegrationFeatures" function. The datasets were then prepared for integration using the "PrepSCTIntegration" function, dimensionality reduction was performed for all datasets using the "RunPCA" command and integration anchors

were established using the "FindIntegrationAnchors" function, with RPCA reduction using the first 30 dimensions and the "k.anchor" argument set to 10. All datasets were then integrated using the "IntegrateData" function, generating a single integrated, batch-corrected expression matrix, which was used for all further downstream analyses.

## Clustering and subclustering of cell types

The Seurat function "RunPCA" was used for principal component analysis (PCA) followed by UMAP (Uniform Manifold Approximation and Projection) dimensionality reduction and Louvain clustering, using the "RunUMAP" "FindNeighbors" and "FindClusters" functions. For sub-clustering analysis, the clusters of interest were subset, split by sample and normalization, variance stabilization and integration was reiterated with the same parameters as described above. Thereafter PCA, UMAP dimensionality reduction and Louvain clustering were reiterated on the reintegrated and pre-processed subset to derive subclusters.

## Differential gene expression analysis

The MAST statistical framework[90] within Seurat's "FindAllMarkers" and "FindMarkers" functions was used for differentially expressed gene (DEG) calculations to identify cluster markers, and between group differences in gene expression, as previously described[74], with minor modifications. Briefly, only genes expressed in a minimum of 10% of nuclei in either tested group were considered. Log-normalized RNA-counts were used for DEG analyses. The number of UMIs and the percentage of mitochondrial reads, were passed to the "latent.vars" argument. For between group comparison we defined a $|\log_2$ fold change $\geq 0.6|$ and Bonferroni-adjusted $p$-value <0.05 as DEG thresholds.

## Module score calculations

Seurat's "AddModuleScore" function was used to calculate module scores, for previously published gene sets, for each nucleus. All gene sets used are described in detail in Supplementary Data 2. Human and mouse gene symbols were converted to human orthologs using the gorth tool in gprofiler2[91], before module score calculation. Estimation of cell cycle phases was conducted using Seurat's "CellCycleScoring" function, as per developer's vignettes.

## Enrichment analysis

Enrichment analysis was performed as previously described[74]. Briefly, rat gene names of DEGs of interest were first converted to human orthologs using the gorth tool in gprofiler2 and used as input for Enrichr[92]. For enrichment analyses of infarction-restricted oligodendrocyte lineage clusters, DEGs derived from the comparison of OPC_1 to OPC_0 and MOLIGO_1 to MOLIGO_2 were imputed. For enrichment analyses of infarction-restricted astrocyte subclusters, DEGs derived from the separate comparison of AC_3, AC_4, or AC_5 to AC_1 and AC_2, pooled were used as input. We queried the gene set databases "GO Biological Process 2023", "GO Molecular Function 2023", "Reactome 2022" and "KEGG 2021 Human". All enriched terms derived from these analyses are reported in Supplementary Data 5 and 10 for oligodendrocyte lineage and astrocyte sub-sclusters, respectively. Only enriched terms with Benjamini-Hochberg method adjusted $p$ values of <0.05 were retained, for further analyses. Significantly enriched terms were then ranked by their respective "Combined Score", which accounts for Fisher's exact test $p$-values and the term's deviation to its expected rank[92]. Up to TOP 20 enriched terms per subcluster and queried database, ranked by "Combined Score" are presented as dot-plots, corresponding gene by term matrices are given in Supplementary Data 5 and 10. The majority of the highlighted genes within Figs. 2 and 4, were derived from these data.

## Cell trajectory based pseudotime inference analysis

We conducted pseudotime trajectory analyses on the oligodendrocyte lineage subset using Monocle3 v.1.3.1[93,94], following the developer's vignettes. First we ordered oligodendrocyte lineage cells along a pseudo-temporal trajectory, to infer whether the observed subclusters followed the established developmental trajectory of the oligodendrocyte lineage. Furthermore, this analysis was conducted to investigate the emergence of stroke-specific deviations from the homeostatic oligodendrocyte lineage trajectory. To this end, we converted the fully processed Seurat subset into a CDS object using the "as.cell_data_set" command and pre-processed the CDC object for subsequent analyses using the "estimate_size_factors" and "preprocess_cds" functions at default parameters and transferred the cell cluster annotations and UMAP cell embeddings from the original Seurat object. Trajectory graph construction and estimation of pseudotime was performed using the "learn_graph" and "order_cells" functions. Because OPCs are the canonical progenitor population of oligodendrocytes and the OPC cluster OPC_0 was the only OPC cluster conserved across all dataset this cluster was chosen as root, for ordering nuclei in pseudotime, which is a measure of the progress of an individual cell through an inferred trajectory.

We also investigated which genes change dynamically across the developmental trajectory and whether these genes could be grouped into co-regulated modules. To achieve this, the Moran's I test based function "graph_test" was used to identify genes, which expressions are correlated or anticorrelated in adjacent cells along the inferred pseudotime trajectory, that is genes which expression changes as a function of pseudotime. "Principal_graph" was passed to the neighbor_graph argument in the function, as indicated by the packages developers and the obtained dataframe was subset to genes with corresponding q-values <0.05 and morans I > 0.05. Thereafter, we used the "find_gene_modles" function, which runs UMAP and subsequent Louvain community analyses to identify co-regulated gene modules, at a resolution of 0.01. For plotting, the aggregate gene expression of all genes within a respective model was generated using the "aggregate_gene_expression" function. All co-regulated gene modules inferred from this analyses are provided in Supplementary Data 6.

## Inference of transcription factor activity

We used the R package decoupleR, as per developers vignettes to infer transcription factor (TF) activities, which estimate the likelihood of the transcriptional activity of a respective TF, based on the expression of its known downstream transcripts within the dataset[41]. Briefly, CollecTRIs´ rat regulon database was retrieved via Omnipath[95] using the "get_collectri" function. DecoupleR´s Univariate Linear Model (ulm) was run on normalized log-transformed RNA counts using the "run_ulm" function, to infer transcription factor activity scores for each nucleus. Inferred transcription factor activity scores were then aggregate for each cluster within each group and presented as heatmaps.

## Cell-cell communication inference analysis

To infer potential cell-cell communication (CCC) events between cell types we used the LIgand-receptor ANalysis frAmework (LIANA) v.0.1.12, following developer's vignettes[95,96]. Briefly, LIANA combines multiple algorithms and underlying databases with known cell-cell communication pairs to infer CCC events, including ligand receptors pairs, but also communication events through secreted enzymes, extracellular matrix proteins, or direct contact (e.g. cell-cell adhesion proteins), based on the gene expressional profiles within the dataset. Using the "generate_homologs" function LIANA's consensus CCC resource entries were converted to *rattus norvegicus* ortholog gene symbols. The functions "liana_wrap" and "liana aggregate" were used at

default settings to infer ligand receptor pairs and obtain consensus ranks across all default CCC methods using Robust Rank Aggregation (RRA). Only predicted ligand receptor interactions with aggregate rank scores ≤0.05 were retained for subsequent analyses.

## Visualization of bioinformatics data

The following R packages were used for data visualization: Seurat v.4.3.0, Monocle3 v.1.3.1, ggplot2 v.3.2.2[97], EnhancedVolcano v.1.16.0[98], UpSetR v.1.4.0[99], scCustomize v.1.1.1[100], SCPubr v. 2.0.1[101], Complex-Heatmap v. 2.14.0[102] and pheatmap v. 1.0.12[103]. For schematics templates from BioRender.com and schematics drawn in Sketchbook v.5.1 (Autodesk Inc., San Rafael, CA, USA), Affinity Designer v.1.10.6.1665 and Affinity Photo v.1.10.5.1342 (Serif, Nottingham, United Kingdom) were used.

## Human brain tissue samples

Archived formalin-fixed, paraffin-embedded (FFPE) biopsy samples from 4 patients, (1 male, 3 females, 33 to 60 years of age) were included. Samples were graded by trained neuropathologists as cerebral infarctions in the stage of macrophage resorption (Stage II) and pseudo cystic cavity formation (Stage III), in accordance with previously described histopathological classifications[104].

## Immunofluorescence staining

For Immunofluorescence (IF) staining 5 μm thick rat coronal whole brain sections and 3 μm thick human FFPE tissue sections were cut from paraffin blocks. After deparaffinization, sections were blocked in 0, 9% $H_2O_2$ in methanol for 10 min and washed three times in ddH₂O, followed by 40 min of heat induced epitope retrieval (HIER) using DAKO Target Retrieval Solution pH6, or pH9 (cat# S2369, S2367, DAKO - Agilent Technologies), in a Braun household food steamer. Sections were allowed to cool for 20 min at room temperature, washed thrice in DPBS, and incubated with 1% sodium borohydride (cat# 1063710100, Merck Millipore, Burlington, MA, USA) in DPBS for 3 minutes to quench autofluorescence, followed by 3 washes in ddH2O and 3 washes in DPBS. Sections were then blocked and permeabilized in protein-blocking buffer (DPBS with 2% BSA, 10% fish gelatin (cat#: G7041, Sigma-Aldrich), 0.2% Triton-X (cat# T9284, Sigma-Aldrich)) for 30 min at room temperature. For staining's in which mouse-derived primary antibodies were applied on mouse tissue, the sections were blocked for an additional 30 min with 10% M.O.M.® (Mouse on Mouse) Blocking Reagent (cat# MKB-2213-1, Vector Laboratories, Newark, CA, USA). For some staining's we directly labelled primary antibodies using FlexAble CoraLite® Plus Antibody Labeling Kits (Proteintech, Rosemont, IL, USA), as per the manufacturer's instructions. To colocalize antigens in tissue sections using primary antibodies derived from the same host species (all rabbit derived) we used the following approach. Tissue sections were incubated with the first primary antibody for 18 h at 4 °C, washed three times in DPBS and incubated with an appropriate secondary antibody for 1 h at room temperature. Thereafter sections were washed three times in DPBS and blocked with 10% rabbit serum in DPBS for 45 min, to block residual unbound epitopes of the secondary anti-rabbit antibodies. Sections were then incubated with fluorophore labeled primary antibodies for 16-18 h at 4 °C, washed thrice in DPBS, incubated with DAPI (cat#: 62248, ThermoFisher Scientific), at a dilution of 1:1000 for 5 minutes, washed again 3 times in DPBS and 2 times in ddH₂O and finally mounted in Aqua Polymount medium (cat#: 18606, Polysciences, Warrington, PA, USA). For immunofluorescence assays using antibodies from different host species, all primary antibodies were applied concomitantly for 16–18 h at 4 °C, sections were rinsed thrice in DPBS, incubated with appropriate secondary antibodies for 1 h at room temperature, washed, DAPI counterstained and mounted as described above. 2% BSA and 5% fish gelatin in DPBS was used as antibody diluent in all assays. All antibodies and labelling kits

used are summarized in Supplementary Table 2 and antibody combinations, dilutions and corresponding HIER treatments, for all IF stainings are detailed in Supplementary Table 3.

## Luxol Fast Blue staining

Myelin was visualized using Luxol Fast Blue (LFB) staining. Briefly, deparaffinized 5 μm thick rat or mouse coronal brain sections were incubated in 0.1% LFB (cat#: 315390-0025, RAL Diagnostics, Martillac, France.) at 56-60 °C over night, rinsed in ddH₂O and incubated for 5 min in 0.1% $Li_2CO_3$ in ddH₂O. LFB staining was developed in 70% ethanol, sections were thereafter rinsed in ddH₂O, and counterstained with Nuclear Fast Red (cat#:10264.01000, Morphisto, Offenbach am Main, Germany) for 20 min, rinsed in ddH₂O and embedded.

## Microscopy and quantification

Sections were imaged at an OLYMPUS BX63 fluorescence microscope, with motorized stage, using Olympus cellSens software (Olympus, Shinjuku, Tokyo, Japan). Tissue sections were scanned at 20x magnification using cellSens´ manual panoramic imagining (MIA) function, with automatic shading correction, at default settings. All downstream analyses were performed in QuPath[105]. Cell counts were obtained in perilesional cortical grey matter and white matter regions of the ipsilateral stroke lesioned hemisphere, as well as anatomically corresponding regions in the contralateral hemisphere and matched section from Sham-operated animals. T2 weighted MRI images from the same animals were used to guide the definition of perilesional and lesional areas. Grey matter (GM) ROIs were defined as 1 mm² (800 × 1250 μm) rectangles, at the border of the stroke lesion. GM ROIs were placed in the perilesional zone, so that the area corresponding to the rim of the T2-weighted hyperintense lesions was located in the middle of the GM ROI rectangle. Due to the variable area and contribution of large white matter tracts to the perilesional and lesion area, white matter ROIs of approximately 1 mm² were defined using QuPaths brush annotation tool, encompassing the corpus callosum and variable portions of the external capsule. In rat brain tissue rectangular core lesion ROIs measuring approximately 2.5–4 mm², depending on the overall size of the lesion were placed in the middle of the lesioned area, as defined by T2-weighted MRI data, in a distance of at least 500 μm from the nearest lesion boarder. In mice rectangular core lesion ROIs of approximately 0.6-0.8 mm², depending on the overall size of the lesion were placed in the middle of the lesioned area, as defined by T2-weighted MRI data in a maximal distance from the lesion boarders.

Cells were identified using the Cell detection function, based on nuclear DAPI signal and intensity features, including Haralick features, as well as smoothed features (Radius(FWHM) = 50 μm) were computed for each channel of interest on every analysed tissue section. For standardized annotation of immunopositive cells, object classifiers were trained, using QuPaths´ random trees algorithm on at least 100 cells per tissue section, for each channel. The obtained cell counts were exported and normalized to 1 mm², for statistical analyses.

## Purification of rodent oligodendrocyte precursor cells OPC

Primary rodent OPCs were purified using differential detachment as previously described[74], with minor modifications. Briefly, forebrains from a total of 20 E20 fetal rat cortices, derived from three timed pregnant Sprague Dawley rat dams (Charles river) were separated from meninges, dissected in ice cold HBSS (cat#: 14175095, Gibco, ThermoFisher Scientific) and enzymatically dissociated using Miltenyis Neural Tissue Dissociation Kit (P) (cat #: 130-092-628, Miltenyi) and a gentleMACS™ Octo Dissociator with Heaters (cat#: 130-096-427,

Miltenyi) (program: 37C_NTDK_1), as per manufacturer's instructions. Ice cold DMEM/F12 + Glutamax 4 mM (cat#: 31331093, Gibco, ThermoFisher Scientific), supplemented with 10% heat inactivated fetal bovine serum (FBS) (cat#: 10500064, Gibco, ThermoFisher Scientific) was used to stop enzymatic dissociation and the cell suspension was filtered (70 µm filters) and centrifuged for 4 min at 300 g, at room temperature, on a swing bucket centrifuge. After decanting the supernatant, the cell pellet was suspended in mixed neural culture medium: DMEM/F12+ Glutamax 4 mM, 10% FBS, 1% Penicillin-Streptomycin (P/S) (cat#: 15140122, Gibco, ThermoFisher Scientific), 1% B27 supplement (cat#: 17504044, Gibco, ThermoFisher Scientific). Cells were seeded in Poly-L-lysin–hydrobromid (PLL) (cat#: P1524, Sigma Aldrich) coated T75 flasks (cat#: CLS430641U, Corning) at a density of approximately 3.5 ×10⁶ cells per T75 flask and maintained in mixed neural culture medium for 8-10 days, with media half changes every 48 h. At day in vitro (DIV) 8-10 flasks were sealed air tight and shaken at 275 rpm, 37 °C on an orbital shaker (MTS 4, IKA-Werke GmbH & Co. KG, Staufen, Germany) in a humidified incubator for 1 h. This step detached the majority of loosely attached microglia, which were removed by a full media change with mixed neural culture medium. Thereafter, the T-75 flasks were allowed to equilibrate in a humidified incubator at 95%O$_2$/5%CO$_2$ for 2 h, resealed and shaken at 300 rpm, 37 °C for 16-18 h. Supernatant with detached OPCs was collected, filtered (40 µm filters) and plated in 94/1 mm non-cell culture treated petri dishes (cat#: 632181, Greiner bio-one) and incubated for 50 min in a humidified cell culture incubator. Supernatant with non-attached OPCs was collected, plates with attached residual microglia were discarded. OPCs containing supernatant was centrifuged at 300 g for 4 min and OPCs were resuspended in defined serum free OPC base medium. OPC base medium consisted of DMEM/F12 + Glutamax 4 mM, 1 mM sodium pyruvate (cat#: 11360070, Gibco, ThermoFisher Scientific), 10 ng/ml d-Biotin (cat#: B4639), 5 µg/ml N-Acetyl-L-cysteine (cat#: A9165), 62.5 ng/ml progesterone (cat#: P8783), 5 µg/ml Insulin (cat#: I6634), 40 ng/ml sodium selenite (cat#: S5261), 100 µg/ml Transferrin (cat#: T1147), 100 µg/ml BSA, all from Sigma Aldrich, 16 µg/ml putrescine (cat#: A18312, ThermoFisher Scientific), 1% P/S and 2% B27.

### In vitro OPC migration assay
2 well culture-inserts (cat#: 80209, Ibidi, Gräfelfing, Germany), in PLL coated 4 well chamber slides (cat#: 354114, Falcon, Corning) were used for migration assays. 50 µl OPC cell suspension at a concentration of 1 × 10⁶ cells/ml were seeded in OPC base medium, supplemented with 20 ng/ml platelet-derived growth factor A (PDGF-A) (cat#: PPT-100-13A-50, Biozol, Eching, Germany), in each well. Cells were allowed to attach for 16–18 h. Thereafter, the culture-insert was removed leaving a defined 500 µm cell free gap. Medium was then changed to 500 µl OPC base medium (untreated control ( = UC)) or OPC base medium supplemented with 1 ug/ml Osteopontin (cat#: 6359-OP, R&D Systems, Minneapolis, MN, USA) (OPN). After 48 h of migration cells were fixed with 4% paraformaldehyde in DPBS for 20 min at room temperature and washed three times with DPBS, followed by blocking and permeabilization in 2% BSA, 10% fish gelatin and 0.2% Triton-X. Cy3® conjugated anti-NG2, diluted 1:50 (cat#: AB5320C3, Sigma Aldrich) and CoraLite® Plus 488 conjugated Ki67, at a concentration of 2 µg/ml (cat#: Ab15589, Abcam, labelled with the FlexAble CoraLite® Plus 488 Antibody Labeling Kit Cat#: KFA001, Proteintech) were used to visualize OPCs and mitosis committed nuclei, respectively. 2% BSA and 5% fish gelatine in DPBS was used as antibody diluent and antibody dilutions were applied over night at 4 °C. Thereafter cells were washed three times in DPBS, incubated with DAPI (1:1000) for 5 min, washed an additional three times in DPBS, and two times in ddH$_2$O before mounting in Aqua Polymount medium. All NG2 positive and NG2/Ki67 double positive cells within the 500 µm gap area, of each replicate, were counted at a OLYMPUS

BX63 fluorescence microscope using Olympus cellSens software (Olympus, Shinjuku, Tokyo, Japan).

### Bromodeoxyuridine (BrdU) incorporation assay
BrdU incorporation assays were used to assess OPC proliferation in vitro. OPCs were plated on PLL coated cover slips (cat#: CB00120RA020MNZ0, Epredia, Portsmouth NH, USA), in 24 well plates (cat#: 3527, Costar, Corning) at a density of 0.5 × 10⁵ cells per well in OPC base medium, supplemented with 20 ng/ml PDGF-A and were allowed to attach and equilibrate for 24 h. Thereafter, PDGF-A supplemented medium was removed and cells were rinsed once in OPC base medium to remove residual PDGF-A. OPCs were then treated with osteopontin at a concentration of 1 µg/ml (OPN condition) untreated OPCs in OPC base medium alone served as controls (UC condition). Cells were treated for 24 h, during the last 6 h 10 µM BrdU (cat# 51-2420KC) was added. Cells were fixed and permeabilized with BD Cytofix/Cytoperm buffer (cat# 51-2090KE) for 20 min, washed thrice in BD Perm/Wash buffer (cat# 51-2091KE) and refixed for an additional 10 min in BD Cytofix/Cytoperm buffer, followed by incubation with 300 µg/ml DNAse (cat# 51-2358KC) in DPBS at 37 °C for 1 h to expose nuclear BrdU, as per manufacturers recommendations. All reagents from BD Bioscience (Franklin Lakes, NJ, USA).

Cells were then washed thrice in BD Perm/Wash buffer, blocked with 2% BSA, 10% fish gelatine in DPBS for 30 min and FITC conjugated anti-BrdU antibody (cat# 51-2356KC, BD Bioscience), and Cy3® conjugated anti-NG2 (cat# AB5320C3, Sigma Aldrich), both diluted 1:50 in 2% BSA and 5% fish gelatine in DPBS were applied. After overnight incubation, cells were washed thrice in DPBS, incubated with DAPI (1:1000) for 5 min, washed an additional 3 times in DPBS, and 2 times in ddH$_2$O before mounting in Aqua Polymount medium. For each condition 4 cover slips were imaged and NG2 positive and NG2/BrdU double positive cells in 2 random 20X magnification fields of view per cover slip were counted at an OLYMPUS BX63 fluorescence microscope using Olympus cellSens software.

### Statistical analyses
For cell counts from IF stainings´ in tissue sections we performed Kruskal–Wallis-H-tests, followed by Dunn's post-hoc comparisons, when comparing more than 2 groups, and Mann–Whitney $U$ tests, when comparing 2 groups, as the data structure did not satisfy the prerequisites for parametric tests. Cell counts obtained from cell culture assays were analysed using unpaired two-sided Student´s $t$ tests. Cell counts are reported as mean ± SD through the main text and represented as box plots, depicting medians, 25th to 75th percentiles as hinges, minimal and maximal values as whiskers, and individual counts as dots throughout all respective figures. A $p$-value of <0.05 was set as threshold for statistical significance. All statistical analyses were carried out using GraphPad Prism v.9.0.0 (GraphPad Software).

### Reporting summary
Further information on research design is available in the Nature Portfolio Reporting Summary linked to this article.

## Data availability
Single nucleus RNA-seq datasets generated in this study have been deposited in the NCBI-GEO database under the GEO accession number: GSE250245. The source data generated in this study underlying all reported figures are provided in the Supplementary and Source Data files. Source data are provided with this paper.

## Code availability
No original custom code was developed within this work. Only previously published and open-access software was used following

developer's instructions, cited and described in detail under the relevant method subsections and supplementary notes.

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

## Acknowledgements

The authors would like to thank Hans Peter Haselsteiner and the CRISCAR Familienstiftung for their ongoing support of the Medical University/Aposcience AG public private partnership aiming to augment basic and translational clinical research in Austria/Europe. The authors acknowledge the core facilities of the Medical University of Vienna, a member of Vienna Life Science Instruments. We also thank Matthias Wielscher for his support in bioinformatics analyses and Irene Erber, Anja Schirin Novak and the team of Matthias Farlik-Födinger for their invaluable technical support. This work was supported by the Austrian Research Promotion Agency (#852748, #862068) and the Vienna Business Agency (#2343727) awarded to HJA. MM was funded by Aposcience AG. This study was also supported by VASCage—Centre on Clinical Stroke Research. VASCage is a COMET Centre within the Competence Centers for Excellent Technologies (COMET) programme and funded by the Federal Ministry for Climate Action, Environment, Energy, Mobility, Innovation and Technology, the Federal Ministry of Labour and Economy, and the federal states of Tyrol, Salzburg and Vienna. COMET is managed by the Austrian Research Promotion Agency (Österreichische Forschungsförderungsgesellschaft)—FFG Project number: 898252. SH gratefully acknowledges funding by the National Institutes of Health (1R01NS114227-01A1 to SH). SzN and GBB were supported by the Austrian Science Fund (FWF), grant numbers T109 awarded to SzN, P31085-B26 awarded to GBB.

## Author contributions

D.B. and M.M.: conceptualization—overall design of the study. D.B., M.K., C.O., S.z.N., G.B.B., S.K., H.J.A., M.M.: conceptualization/methodology—design of in vivo experiments, sample acquisition and general analysis strategy. C.O., A.L., A.C.: investigation—execution of in vivo MCAO and MRI experiments. D.B., D.C., K.K., M.D.: investigation—scRNAseq experiments. D.B., E.P., C.J.R., G.T., P.J., B.G., H.K., M.S., C.H., V.E., N.S.: investigation—immunofluorescence staining experiments. D.B., C.J.R., G.T., H.K., M.S., N.S.: investigation—cell culture experiments. D.B., M.K., C.O., A.L., A.C., S.z.N., G.B.B., S.K.: formal analysis/data curation—MRI data. D.B., E.P., C.J.R., P.J., S.H., R.H., M.M., N.S.: formal analysis/data curation—immunofluorescence staining and cell culture data. D.B.: formal analysis and data curation—bioinformatics analyses. D.B., D.C., K.K., M.D.: data curation/validation—bioinformatics analyses. H.J.A., M.M.: funding acquisition. H.J.A., M.M.: project administration. C.O., S.H., S.K., R.H., H.J.A., M.M.: resources. R.H., S.H., H.J.A., M.M.: supervision—H.J.A. and M.M. supervised D.B., R.H., and S.H. supervised C.J.R, G.T., V.E., C.H. D.B., and M.M.: writing—original draft, visualization. All authors were involved in review & editing and approved of the final manuscript.

## Competing interests

The authors declare no competing interests.

## Additional information

[1]Applied Immunology Laboratory, Department of Thoracic Surgery, Medical University of Vienna, 1090 Vienna, Austria. [2]Aposcience AG, 1200 Vienna, Austria. [3]Department of Neurology, Medical University of Innsbruck, Anichstraße 35, 6020 Innsbruck, Austria. [4]VASCage, Centre on Clinical Stroke Research, 6020 Innsbruck, Austria. [5]Department of Dermatology, Medical University of Vienna, 1090 Vienna, Austria. [6]Division of Neuropathology and Neurochemistry, Department of Neurology, Medical University of Vienna, 1090 Vienna, Austria. [7]Comprehensive Center for Clinical Neurosciences and Mental Health, Medical University of Vienna, Vienna, Austria. [8]Normandie University, UNICAEN, ESR3P, INSERM UMR-S U1237, Physiopathology and Imaging of Neurological Disorders (PhIND), Institut Blood and Brain @ Caen-Normandie (BB@C), GIP Cyceron, Caen, France. [9]Department of Clinical Research, Caen-Normandie University Hospital, Caen, France. [10]Center for Medical Physics and Biomedical Engineering, Medical University of Vienna, 1090 Vienna, Austria. [11]Division of Nephrology and Dialysis, Department of Internal Medicine III, Medical University of Vienna, 1090 Vienna, Austria. [12]Department of Orthopedics and Trauma Surgery, Medical University of Vienna, 1090 Vienna, Austria. [13]Institute of Neurobiochemistry, CCB-Biocenter, Medical University of Innsbruck, 6020 Innsbruck, Austria. [14]These authors contributed equally: Hendrik J. Ankersmit, Michael Mildner. ✉e-mail: hendrik.ankersmit@meduniwien.ac.at; michael.mildner@meduniwien.ac.at

