## [Peer Review File · Nature Communications]

Reviewers' Comments:

Reviewer #1:

Remarks to the Author:

This manuscript describes transcriptomic and functional alterations in neuroglia after experimental ischemic stroke in rats, with a special focus on proliferative oligodendrocyte precursor cells (OPCs), mature oligodendrocytes and astrocytes. The authors describe shared altered pathways and functional alterations in neuroglia in response to cell-cell communication with myeloid cells after stroke, which are regulated in an osteopontin-dependent manner, and validate candidate genes at the protein level using immunofluorescence stainings. Specifically, OPCs show increased migratory capacity, when stimulated with osteopontin in vitro. Overall, the manuscript offers valuable insights in neuroglial responses to post-ischemic changes in the brain and describes a potential pathway via osteopontin, which regulates the described alterations in neuroglial reactivity after stroke. Despite the magnitude of recently published transcriptomic studies, the present manuscript demonstrates novel data of transcriptomic changes in OPCs, oligodendrocytes and astrocytes and highlights the commonalities of injury response based on cell-cell communication with myeloid cells after stroke. Therefore, the manuscript is an enrichment to the experimental stroke field.

Main points:

- 1) The used dataset is limited to only one timepoint after stroke. Compared with existing post-stroke snRNA or scRNA sequencing datasets, the presented data does not give knowledge about manifestations of glial responses after stroke or allow for dynamic evaluation of post-stroke glial responses, which would be highly advantageous.
- 2) Merging of mMCAO and sMCAO snRNA seq datasets: the two stroke severity models affect different structural and functional brain areas, and thus, different cell populations. Although computational integration methods may allow for sufficient coverage of the datasets, there might be a "harmonization bias" due to data adaptation. As seen in Fig. 1 d), the relative cluster abundance in the ipsilateral data from mMCAO and sMCAO is very different. It would be good to show, that the relevant clusters and/or celltype variations used for the further analysis steps are comparable between the two ipsilateral datasets.
- 3) The expression analysis of curated genes (Fig. 2d), Fig. 4c-e) seems rather subjective. It would be better to show functional association of transcriptomic alterations using unbiased and objective methods, such as gene set enrichment analysis or pathway analysis, which is dependent on differentially regulated genes between the analyzed treatment groups.
- 4) Although astrocytes are one of the main glial type analyzed in this manuscript, the astrocyte population is very limited among the entire dataset (only <2 % of total isolated nuclei)
- 5) It remains unclear how the link to osteopontin dependent glial response regulation was created.
- 6) Microglia cluster was not definitely labeled as such, even though there are widely accepted transcriptomic markers available (Hexb, P2ry12, Tmem119) and might also functionally be distinct from post-stroke infiltrated myeloid cells from the circulation. Thus, it would be important for the later link of myeloid cells being the source of osteopontin and regulating the migratory function of astrocytes and oligodendrocytes, to differentiate if microglia or infiltrated monocytes are the source of osteopontin.
- 7) The sham and contra samples of the dataset contain hardly any myeloid cells, which includes microglial cells. These should be of higher abundance since microglia account for 10-15 % of brain cells, although their population size is increased in the ipsilateral hemisphere after stroke (Fig. 1b, Suppl. Fig. 2b).
- 8) Suppl Fig 14 – where is osteopontin labelled in the images?

Minor points:

- 1) Avoid red/green coloring within the same figures to ensure readability for readers with colorblindness
- 2) OPC migration assay: only one experiment was performed. A validation of this experiment with

a second experiment would show reliability of the assay (at least 2 biological replicates, at least two independent experiments, best case biological replicates in doublets).

Reviewer #2:

Remarks to the Author:

Bormann et al have done single nucleus RNA sequencing (snRNAseq) analyses in a rat model for stroke (MCAO). They performed MCAO and sham surgeries on male rats. Injury size was documented by MRI 48 hours after the induced ischemia and cells were harvested thereafter. The authors identified several transcriptional changes associated with the stroke. Interestingly, the most striking changes were in astrocytes and oligodendrocytes. They identified osteopontin to CD44 signaling as one of the most robust immunoglia interaction events, and they then used immunofluorescence staining to confirm osteopontin expression and its effect on OPC migration.

This is an important study because it gives insights into the cell types and pathways that are altered in stroke. However, the study has some important limitations that need to be addressed (listed below). I also think the manuscript needs a lot of editing to improve readability of text and figures. Per now, the manuscript is not accessible to people that are not experts in snRNAseq analyses.

Major concerns:

- 1) The snRNAseq was performed on coronal brain slices that were just cut in half (to separate l/r hemisphere). This approach does not discriminate between peri-infarct (perilesional) area and infarct core in the snRNAseq. Nor does it discriminate between different relevant brain regions such as neocortex, striatum or hippocampus. Redoing this part with new snRNAseq would probably be too demanding, but some more analyses should be done by immunostaining (and/or other methods such as western blotting) to confirm snRNAseq findings and to determine in which regions the ischemia-associated changes occur. For instance, are the ischemia-related cell types typically in the ischemic core or in the peri-infarct area?
- 2) Immunofluorescence analyses in Figs. 3, 5 and 6 claim to be in the perilesional zone, but it is not clear whether they are looking at sections from moderate or severe MCAO or both. My impression is that the MRI images in those figures are from severe MCAO where the indicated area seems to be part of the infarct core. Analyses should be done in confirmed perilesional areas and moderate and severe MCAO should probably be separated. A better explanation of how they define the perilesional zone may also help here.
- 3) Although the English language and grammar is fine, the manuscript and the figures are a very heavy read. The figures are extremely busy, with several panels (often four) under each letter. The figures should be simplified by removing non-necessary panels (around 50% can be removed or transferred to supplement). The Results main text is also hard to follow. It should be simplified, and details can be moved to Methods or Supplement.
- 4) Fig 1D: The Olig2 group is much smaller in Sham R than in Sham L. I would have expected these to be similar. Needs to be discussed.
- 5) Cells were only sampled at one time point (48h). This is an important drawback and should be discussed more thoroughly.
- 6) The authors mention in the Discussion that some of the astrocyte results are "limited by the low number of captured AC_5 nuclei". However, it is not clear from the Results section that AC_nuclei was low. Does this mean AC_5 results are less reliable?

Reviewer #3:

Remarks to the Author:

I co-reviewed this manuscript with one of the reviewers who provided the listed reports. This is part of the Nature Communications initiative to facilitate training in peer review and to provide appropriate recognition for Early Career Researchers who co-review manuscripts

Reviewer #4:

Remarks to the Author:

The paper by Daniel Bormann et al., titled "Single Nucleus RNA Sequencing Reveals Glial Cell Type-Specific Responses to Ischemic Stroke," explores the cellular landscape and microenvironment during the early stages of ischemic stroke recovery (48 hpi) in a rat stroke model. Utilizing snRNA-seq, the study uncovers molecular signatures in microglia, oligodendrocyte lineage cells, and astrocytes, elucidating their transcriptional perturbations during acute recovery. Notably, the authors characterize infarction-restricted proliferating oligodendrocyte precursor cells (OPCs), mature oligodendrocytes, and reactive astrocyte populations, revealing commonalities in their response to ischemic injury. The paper also explores immuno-glial cross talk with stroke-specific myeloid cells, particularly the involvement of osteopontin-positive myeloid cells and their interaction with CD44-positive OPCs and astrocytes. Finally, the in vitro experiment demonstrates the functional relevance of osteopontin in enhancing OPC migratory capacity.

The manuscript is well-written and provides a thorough analysis of the snRNA-seq data, offering valuable insights into cellular responses following ischemic stroke. The authors' approach in characterizing infarction-specific molecular signatures in oligodendrocyte lineage cells and astrocytes is commendable, documented meticulously in numerous figures and supplementary tables. The inclusion of immunohistochemistry to validate stroke-induced populations adds a robust experimental layer, reinforcing the reliability of the identified cell states. The exploration of cell-cell communication, particularly the shared immuno-glial cross talk between stroke-induced OPCs, astrocytes, and stroke-specific myeloid cells, is a noteworthy aspect of this study. The immunohistochemistry analysis of osteopontin-positive myeloid cells in close proximity to CD44-positive OPCs and reactive astrocytes within the perilesional zone, along with subsequent functional validation showing the enhancement of OPC migratory capacity after osteopontin treatment, adds a functional dimension to the findings.

In summary, I find no major objections to the manuscript and support its acceptance for publication. I have included some minor comments and suggestions for the authors' consideration, which I believe will further enhance the clarity and impact of the study.

- The authors utilized a 5' sequencing approach, which is not standard in the field. Could they provide a rationale for this choice?
- While I concur with the authors' interpretation of the MC_OLIGO cluster, have they considered the possibility that these cells represent doublets?
- Lines 649-650 state, "Pseudotime trajectory analysis indicated that the stroke-specific sub-cluster OPC_1 branched directly from the conserved sub-cluster OPC_0 (Fig.2b)." This claim may not be easily visible in the figure; clarification is suggested.
- Have the authors explored the overlap of the OPC_1 cluster with any "disease-associated" cluster of OPC in the literature? Such a comparison could yield insights into similarities and differences across pathologies.
- Some descriptive paragraphs may benefit from concluding sentences to highlight major findings and improve overall readability.
- The authors might consider to highlight certain discussed ligand-receptors in Supplementary Figure 11, particularly the spp1-CD44 signaling pathway.
- A detailed analysis of human tissues, including quantification and comparison to unaffected

tissue, could strengthen the paper. If revisited, this figure may warrant inclusion in the main text rather than the supplement.

- The suggestion to use the dash symbol "-" consistently for instances where single-cell or single-nucleus is used as an adjective is noted for improved consistency.

Point-by-point response

Reviewer #1 (Remarks to the Author):

This manuscript describes transcriptomic and functional alterations in neuroglia after experimental ischemic stroke in rats, with a special focus on proliferative oligodendrocyte precursor cells (OPCs), mature oligodendrocytes and astrocytes. The authors describe shared altered pathways and functional alterations in neuroglia in response to cell-cell communication with myeloid cells after stroke, which are regulated in an osteopontin-dependent manner, and validate candidate genes at the protein level using immunofluorescence stainings. Specifically, OPCs show increased migratory capacity, when stimulated with osteopontin in vitro. Overall, the manuscript offers valuable insights in neuroglial responses to post-ischemic changes in the brain and describes a potential pathway via osteopontin, which regulates the described alterations in neuroglial reactivity after stroke. Despite the magnitude of recently published transcriptomic studies, the present manuscript demonstrates novel data of transcriptomic changes in OPCs, oligodendrocytes and astrocytes and highlights the commonalities of injury response based on cell-cell communication with myeloid cells after stroke. Therefore, the manuscript is an enrichment to the experimental stroke field.

Response:

We thank the reviewer for the encouraging assessment of our manuscript and the thorough and constructive feedback and address all raised concerns point by point below.

Main points:

1) The used dataset is limited to only one timepoint after stroke. Compared with existing post-stroke snRNA or scRNA sequencing datasets, the presented data does not give knowledge about manifestations of glial responses after stroke or allow for dynamic evaluation of post-stroke glial responses, which would be highly advantageous.

Response:

We fully agree with the reviewer, evaluating transcriptional changes over multiple time points would be highly informative. In light of the comprehensive analysis conducted on our current dataset, comprising 7 primary figures and 33 supplementary figures, we have determined that conducting additional experiments to investigate time kinetics would be excessive and could overshadow our primary findings. Consequently, we have chosen not to include additional experiments addressing this aspect. However, we of course concede that we need to discuss this limitation of our study more diligently and have revised the discussion of our manuscript accordingly.

2) Merging of mMCAO and sMCAO snRNA seq datasets: the two stroke severity models affect different structural and functional brain areas, and thus, different cell populations. Although computational integration methods may allow for sufficient coverage of the datasets, there might be a "harmonization bias" due to data adaptation. As seen in Fig. 1 d), the relative cluster abundance in the ipsilateral data from mMCAO and sMCAO is very different. It would be good to show, that the relevant clusters and/or celltype variations used for the further analysis steps are comparable between the two ipsilateral datasets.

Response:

Indeed, the reviewer highlights an important potential pitfall of the chosen computational integration method. In compliance with the reviewers request we performed several new calculations in the revised version of our manuscript to address this issue. Within the previous version of our manuscript we have not segregated the myeloid cell populations by stroke severity. We have now added a dedicated supplementary figure (Supplementary Fig.5) directly comparing the abundance of the different myeloid cell subsets in moderate and severe strokes, as well as their transcriptional profiles and overlaps with myeloid cell scores in previously described myeloid cell transcriptional phenotypes. Indeed, while the transcriptional profiles of the different myeloid cell sub clusters are largely comparable, these analyses highlighted that macrophage enriched subsets are more abundant in the severely infarcted tissue as compared to moderately infarcted tissue.

Moreover, in the previous version of our manuscript we have only shown that the MCAO ipsi specific sub clusters (OPC_1 and MOLIGO_1) are indeed captured across both stroke severities, although particularly MOLIGO_1 nuclei are more abundant in the severe infarctions.

We fully agree with the reviewer in that it would be important to check whether the relevant sub clusters differ substantially between the two severities before considering further integrated analysis steps. Direct clusters wise comparison of the two ipsilateral datasets however did not reveal substantial transcriptional differences between the two ipsilateral datasets, within the oligodendrocyte lineage subclusters (see: Supplementary Fig. 10d).

We concluded that OPC_1 and MOLIGO_1 are indeed virtually distinctly captured in infarcted tissue, of both moderate and severe infarctions but do not differ transcriptionally between the two ipsilateral datasets, in a meaningful manner. Therefore, we deemed it to be reasonable to integrate the two severities for subsequent analysis. Nevertheless, we also provide an additional supplementary figure showing the expression of the curated OPC_1 and MOLIGO_1 marker DEGs depicted within the main figure (Fig.2i) separately for each sequenced dataset (see Supplementary Fig.14).

Regarding the transcriptional heterogeneity of stroke reactive astrocytes, we indeed have already denoted several differences between the abundance of the different reactive astrocyte population in the moderate and severe infarction derived datasets. Particularly AC_4 and AC_5 were more enriched in the severe as compared to the moderately infarcted tissue. Indeed, the majority of nuclei contained with this cluster are derived from severe infarctions. As the reviewer correctly points out the severe strokes extend to different structural and functional brain areas compared to the more moderate strokes. It is thus possible that the AC_4 and AC_5 cell states might derive from distinct spatially restricted astrocytes, which were affected within severe but not moderate strokes. Alternatively, the emergence of these cell states might be

dependent on the overall lesion size per se. Both of these explanations are mentioned explicitly. Moreover, similar as for the oligodendrocyte lineage subsets, we provide a dedicated new figure (Supplementary Fig. 21) in which the relevant clusters and celltype variations are shown for each sequenced dataset individually.

3) The expression analysis of curated genes (Fig. 2d), Fig. 4c-e) seems rather subjective. It would be better to show functional association of transcriptomic alterations using unbiased and objective methods, such as gene set enrichment analysis or pathway analysis, which is dependent on differentially regulated genes between the analyzed treatment groups.

Response:

This point is well taken and we thank the reviewer for this important remark. The curation of genes for Fig.2d and Fig. 4c-e in the previous manuscript was in fact driven precisely by the results of gene set enrichment and pathway analysis, which we have reported in the Supplementary Data (see Supplementary Data in the revised manuscript). We recognize that our rationale for the selection of genes for the heatmaps in Fig.2 and Fig. 4 cannot be unambiguously deduced from the information given in the original manuscript and therefore provide additional information in the revised methods section, new supplementary figures and new supplementary data.

For gene curation the following approach was chosen:

Differentially expressed genes derived from the comparison of infarction specific cell clusters to their homeostatic counter parts, that is OPC_1 vs OPC_0, MOLIGO_1 vs MOLIGO_2 and AC_3, AC_4, AC_5 vs AC_1 and AC_2 where subset to stringent DEGs, applying a cut-off of a $|\log_2\text{fold change}| \geq 0.6$ and Bonferroni-adjusted p-value < 0.05 , as described in the methods section. The rat gene names of these DEGs of interest were then converted to human orthologs using the gorth tool in gprofiler2 and used as input for Enrichr. We then queried the gene set databases “GO Biological Process 2023”, “GO Molecular Function 2023”, “Reactome 2022” and “KEGG 2021 Human”. We then flagged all enriched terms with Benjamini-Hochberg method adjusted p values of < 0.05 . The unbiased result of these analyses are given in Supplementary Data 5 and 10.

To focus on the most robustly enriched functional annotations we then ranked the enriched terms by their respective “Combined Score”, which combines p-values from Fisher's exact test and the respective term's deviation to its expected rank. We decided to focus on the Top 20 enriched terms, as ranked by descending Combined Scores. In the revised manuscript we now depict these terms as dot plots, to allow the reader to assess the results of enrichment analysis at a glance see Supplementary Figs.11,12,22,23,24.

Next we assessed which DEGs are associated to which of the TOP 20 enriched term using a binary matrix. We have now added this data to the revised supplementary data files (see Supplementary Data 5 and 10). We have consciously decided to present this data as an excel file and not a heatmap or network plot, as we believe it is much easier for the inclined reader to navigate this matrix in an excel format. The vast majority of genes presented in the main heatmaps in Fig.2i and Fig 4.h-j was curated from this data frame. In the main figures we aimed to highlight genes that are associated to multiple functionally interrelated terms and then took the liberty of labelling the genes in the heatmaps with descriptive overarching terms. For

example, among the enriched terms derived from the analysis of DEGs which were upregulated in MOLIGO_1 many were associated to the modulation of neurogenesis (e.g. “Axonogenesis”, “Neuron Projection Guidance”, “Regulation of Axon Extension involved in axon guidance”), or cell-cell and cell-ECM interactions, such as “Cell-Cell Communication”, “Cell Junction Organization”, “Regulation Of Focal Adhesion Assembly”. Likewise, instead of reporting the numerous mitosis and cell cycle progression associated terms associated to OPC_1 upregulated DEGs *verbatim* we labeled selected associated genes with the description “Proliferation”. Moreover, some terms were shortened for example “Positive Regulation of Notch Signaling Pathway” was shortened to “Notch signaling”, in the main figure. This approach was chosen in an effort to condense the vast information derived from gene set enrichment analysis into a more digestible heatmap. Moreover, some genes were selected for the main heatmaps due to associations derived from relevant literature, for example DAO genes in Fig.2. Some curated DEGs such as *Igf1r*, *Wdfy1*, *Ybx3* or *App* did not associate clearly to one of the TOP 20 enriched terms, but ranked among the most robustly and specifically enriched DEGs within stroke specific clusters. Therefore, we decided to highlight them in Fig.2i. Although the selection of the vast majority of the curated genes was driven by the unbiased prediction of gene set enrichment analysis, there remains a subjective element regarding which enriched pathways we choose to highlight, which is why we phrased this approach “curation”. It is entirely possible that some biologically relevant pathways are not highlighted in the main figures, for example regarding the differential expression of certain RHO and RAC GTPase family members, which other research might deem highly important. As the extensiveness of our main figures was already assessed critically by reviewer 2 we have refrained from expanding Fig.2i and 4h-j. However, with the newly added supplementary information the full results of all enrichment analyses are provided to the reader transparently and extensively and can be easily explored further.

4) Although astrocytes are one of the main glial type analyzed in this manuscript, the astrocyte population is very limited among the entire dataset (only <2 % of total isolated nuclei).

Response:

The reviewer mentioned an important drawback of the herein chosen nuclei isolation and snRNAseq approach. Indeed, our dataset evidently underrepresents astrocyte transcripts, a common observation in many snRNAseq datasets derived from CNS tissue, for example discussed in Sadick et al. (2022)¹. We thank the reviewer for explicitly mentioning this limitation and concede that it would be diligent to discuss this limitation more rigorously. Indeed, due to the low number of astrocytes in our dataset it is highly plausible that we do not capture the full *in vivo* transcriptional heterogeneity of reactive astrocytes in infarcted tissue. This limitation of our study is now explicitly mentioned within the revised discussion section of our manuscript.

5) It remains unclear how the link to osteopontin dependent glial response regulation was created.

Response:

We thank the reviewer for pointing out that this crucial point needs further clarification. The link between OPN (encoded by *Spp1*) and CD44 was established as follows. Firstly, we and other groups (e.g. Beuker et al. (2022)²) observed a prominent upregulation of *Spp1* in myeloid cells within infarcted tissue specifically (see Supplementary Figs.4,5). Secondly, we observed an upregulation of a canonical receptor of OPN, that is CD44 in astrocytes (see Fig.4), as expected in infarcted tissue, as well as the unexpected upregulation of CD44 in OPCs. Thirdly, our CCC analysis (see Supplementary Figs. 26) predicted a ligand-receptor based infarction specific communication between myeloid and neuroglial celltypes via this axis. Thirdly, abundant prior research, as cited in the manuscript, established an involvement of OPN - CD44 signaling in chemotactic cell migration. We thus reasoned that if a migration of CD44 positive cells towards OPN occurred *in vivo*, in response to infarction we should be able to observe a spatial colocalization of CD44 and OPN positive cells *in situ*. In the previous version of our manuscript this background information was only mentioned briefly within the description of the rationale for the *in vitro* cell migration assays. We therefore now elaborate on this point earlier in the revised version of our manuscript, to illustrate our reasoning more clearly (see page 19 of our revised manuscript). We indeed observed, that both CD44 positive reactive astrocytes, as well as OPN expressing myeloid cells both accumulated at the lesional borders 48h after MCAO. Note that replication in the thromboembolic MCAO model in mice showed that this finding was independent of species and MCAO model. Abundant prior research also proposed that OPN elicits a migration facilitating effect in various cell types, including astrocytes, as cited within the manuscript, while such an effect was not yet shown for OPCs. Hence we addressed this question using an *in vitro* model. Collectively, these findings thus implicate a possible role of the OPN->CD44 signaling axis in governing the cellular composition of the early ischemic lesion. However, as elaborated in detail in the discussion section we do not propose that this signaling axis is the sole driver of the attraction of neuroglia towards the infarct lesion. For example, we cite multiple studies which implicated hyaluronic acid – CD44 signaling as a driver of neuroglial and myeloid cell migration towards the lesion. Likewise, we discuss the pleiotropic involvement of OPN as well as CD44 in multiple diverse signaling pathways. This was also evident within our CCC-Analysis, as multiple specific ligand receptor interactions involving osteopontin and CD44, apart from *Spp1*->CD44 signaling were predicted.

6) Microglia cluster was not definitely labeled as such, even though there are widely accepted transcriptomic markers available (*Hexb*, *P2ry12*, *Tmem119*) and might also functionally be distinct from post-stroke infiltrated myeloid cells from the circulation. Thus, it would be important for the later link of myeloid cells being the source of osteopontin and regulating the migratory function of astrocytes and oligodendrocytes, to differentiate if microglia or infiltrated monocytes are the source of osteopontin.

Response:

We fully agree with the reviewer on the relevance of the identification of the main sources of osteopontin. Regarding the nomenclature of the myeloid cells within our dataset we made the

conscious decision to term the main immune cell cluster (Fig.1) in our dataset myeloid cells, as our sub clustering analysis revealed that this cluster consist of both unambiguous microglia clusters, but also clusters highly enriched in macrophage markers (see Supplementary Figs.4,5). As shown in Supplementary Fig.4d, *Spp1*, as well as multiple other SAMC marker genes are enriched in the microglia subcluster MG_1, as well as in the macrophage transcript enriched clusters (Mφe_1 to 3). To further illustrate this assessment, we would like to refer the reviewer to Reply Fig.1, which indicates that *Spp1* is enriched in both unambiguous microglial as well as macrophage enriched subclusters, although the relative expression of *Spp1* is slightly higher in the microglia cluster MG_1 compared to the macrophage transcript enriched clusters. These findings are well in line with the data presented by Beuker et al. (2022)². Using several complementary methods, the authors showed in great detail that *Spp1* expressing SAMCs are of mixed resident microglial and peripheral macrophage origin, although resident microglia are more predominant sources of SAMC. Our data does not conflict with these findings, which is why we concurred with their conclusion on that the SAMC phenotype, as a specific response to ischemia is likely instructed by the infarcted tissue on to both peripherally derived macrophages and microglia alike.

Reply Fig.1 Spp1 is enriched in both microglia and macrophage transcript enriched myeloid cell subclusters. **a.** Subclustering of myeloid cells derived from infarcted tissue (MCAO ipsi). UMAP plot depicting 2646 nuclei annotated to 6 sub clusters. The relative contribution of each subcluster to all MCAO ipsi enriched myeloid cells is presented as pie plot. **c-g.** Joint density plots depicting gene-weighted kernel density estimation jointly for Spp1 and the microglia markers Hexb (**c**), Tmem119 (**d**), as well as Apoe (**e**) and the macrophage associated markers Cxcr4 (**g**) and Lyz2 (**f**).

7) The sham and contra samples of the dataset contain hardly any myeloid cells, which includes microglial cells. These should be of higher abundance since microglia account for 10-15 % of brain cells, although their population size is increased in the ipsilateral hemisphere after stroke (Fig. 1b, Suppl. Fig. 2b).

Response:

Indeed, we fully agree that the low abundance of microglia and myeloid cells in the Sham datasets is unfortunate. We propose that the chosen method of nuclei isolation is the most likely cause of this phenomenon.

The herein used nuclei isolation protocol was strongly influenced by the approach described in work by Maitra et al. (2021)³. Nuclei extraction from frozen brain tissue samples using Iodixanol gradient centrifugation has been successfully used by this group and was shown to be particularly well suited to recover neuronal and oligodendrocyte clusters (the latter being one of our goals when we considered different nuclei extraction protocols), as shown in works of the same group (see for example work by Nagy et al. (2020)⁴ – Figure 2). Please note that only 1.68% of the cells recovered in this study were annotated as microglia and myeloid cells. Furthermore, the expression of canonical microglia marker genes such as CX3CR1, although restricted to this clusters is rather sparse as evident in the dotplot in Figure 2b⁴ and only present in few cells of this cluster. This suggests the possibility that not even the full 1.68% of the cells annotated as myeloid cells in this study are actually *bona fide* microglia and other myeloid cells. It is thus reasonable to assume that this general approach of nuclei isolation and snRNAseq with frozen brain tissue as starting material heavily favors the enrichment of neuronal and glial transcripts at the evident expense of microglia and myeloid cells, among other populations. Indeed, in previous work of our group⁵ using the herein described protocol myeloid cells were virtually absent in datasets of CNS tissue from Sham, Saline and LPS treated animals.

The reviewer correctly mentioned that microglia account for 10-15% of brain cells *in vivo*, in homeostatic conditions. Regrettably, every individual nucleus and scRNAseq methodology exhibit certain constraints, resulting in diminished cell quantities or even total absence of particular cell types. Many technical parameters such as different cell size, nuclear size and density, or RNA content contribute to which extend different cell populations are represented in the dataset.

We propose that the vastly increased number of myeloid cells in the infarcted brain tissue, combined with the pronounced loss of vital neuronal and glial cells, increased the probability of myeloid cell nuclei capture. This is particularly evident in the severe MCAO datasets (Fig.1). Although our inability to directly compare stroke enriched, reactive myeloid cells to homeostatic microglia and macrophages within our dataset is regrettable, numerous prior studies, cited in this manuscript (e.g. Beuker et al. (2022)) have conducted these comparisons in great detail. Reassuringly, the previously established phenotype of stroke associated myeloid cells was well conserved in our dataset.

8) Suppl Fig 14 – where is osteopontin labelled in the images?

Response:

We thank the reviewer for making us aware of a labelling mistake in the legend of this figure. In the previous version of the manuscript we have erroneously stated that magenta labels Ki67. Magenta labels OPN in this figure. We apologize for this mistake, the figure label was changed accordingly and we have added labels directly to the figure to avoid confusion. (see Supplementary Fig.33, in the revised manuscript).

Minor points:

1) Avoid red/green coloring within the same figures to ensure readability for readers with colorblindness.

Response:

We thank the reviewer for this important suggestion. We strived to make the figures more assessable to readers with divergent color perception, by introducing the following changes:

1. We fully revised the color codes of our figures using publicly available guidelines⁶ and repositories of colorblind-friendly color palettes: <https://personal.sron.nl/~pault/#sec:qualitative>.
2. In cases where the combined use of some red and green chroma was unavoidable, due to the large number of individual cell clusters, for example in Figure 1. we numbered clusters to provide an unambiguous labelling.

Lastly, we used COBLIS - <https://www.color-blindness.com/coblis-color-blindness-simulator/>, a publicly available color blindness simulator to check whether our revised figures are assessable to people with the most common color vision deficiencies.

2) OPC migration assay: only one experiment was performed. A validation of this experiment with a second experiment would show reliability of the assay (at least 2 biological replicates, at least two independent experiments, best case biological replicates in doublets).

Response:

We fully agree with the reviewer on the importance of replication to show assay reliability, and replicated the reported *in vitro* assays.

The number of migrated NG2+ cells was now assessed in 3 independent experiments, with 3-4 replicates per experiment. The percentages of Ki67 positive cells within the cell gap was assessed in 2 independent experiments with 3-4 independent replicates per experiment. The BrdU assay was performed twice independently, assessing the percentage of BrdU positive cells in 3-4 replicates per experiment. For each independent experiment a separate preparation of mixed cultures and isolation of OPCs from an independent litter was performed. Each replicate represents an independent cover slip or 2 well culture-inserts array. All individual counts from all experiments are now reported in the source data file.

Reviewer #2 (Remarks to the Author):

Bormann et al have done single nucleus RNA sequencing (snRNAseq) analyses in a rat model for stroke (MCAO). They performed MCAO and sham surgeries on male rats. Injury size was documented by MRI 48 hours after the induced ischemia and cells were harvested thereafter. The authors identified several transcriptional changes associated with the stroke. Interestingly, the most striking changes were in astrocytes and oligodendrocytes. They identified osteopontin to CD44 signaling as one of the most robust immunogial interaction events, and they then used immunofluorescence staining to confirm osteopontin expression and its effect on OPC migration.

This is an important study because it gives insights into the cell types and pathways that are altered in stroke. However, the study has some important limitations that need to be addressed (listed below). I also think the manuscript needs a lot of editing to improve readability of text and figures. Per now, the manuscript is not accessible to people that are not experts in snRNAseq analyses.

Response:

We are grateful for the reviewer's interest and careful review of our work. Below we address all comments and suggestions.

Major concerns:

1) The snRNAseq was performed on coronal brain slices that were just cut in half (to separate l/r hemisphere). This approach does not discriminate between peri-infarct (perilesional) area and infarct core in the snRNAseq. Nor does it discriminate between different relevant brain regions such as neocortex, striatum or hippocampus. Redoing this part with new snRNAseq would probably be too demanding, but some more analyses should be done by immunostaining (and/or other methods such as western blotting) to confirm snRNAseq findings and to determine in which regions the ischemia-associated changes occur. For instance, are the ischemia-related cell types typically in the ischemic core or in the peri-infarct area?

Response:

In this and the following suggestion the reviewer raises several important interconnected points. We addressed these concerns with novel data in our revised manuscript and jointly discuss these changes in detail below under point 2.

2) Immunofluorescence analyses in Figs. 3, 5 and 6 claim to be in the perilesional zone, but it is not clear whether they are looking at sections from moderate or severe MCAO or both. My impression is that the MRI images in those figures are from severe MCAO where the indicated area seems to be part of the infarct core. Analyses should be done in confirmed perilesional areas and moderate and severe MCAO should probably be separated. A better explanation of how they define the perilesional zone may also help here.

Response:

The definition of perilesional areas was guided by the identification of hyperintense lesions on T2-weighted MRI data, which we have obtained for all animals included in dataset. Therefore,

we were able to estimate the extend of the lesion on IF images using MRI data. Although this approach is well established in previous work (e.g. Bonfanti et al. (2017)⁷), we concede that the correlation of MRI image data to histologically defined hallmarks of the ischemic lesion might not be self-evident. In the new Supplementary Fig.2 we demonstrate that the hyper intense lesion on T2-weighted MRI images corresponds remarkably close to the loss of neurons, as visualized by MAP2 immunofluorescence staining and myelin-loss, as visualized by Luxol Fast Blue staining, in rat brain tissue, 48h post MCAO. Following the reviewers' guidance, we also provide a more precise definition of the regions of interest (ROIs) within the revised methods section.

Regarding the nomenclature of the ROIs the reviewer raises an important point. While in rats the grey matter ROI borders the edge of the lesion in a strict sense, the white matter ROI, that is the corpus callosum and parts of the external capsule are located between the lesioned striatum and lesioned cortex in most cases.

Within the infarcted hemisphere the myelin appears affected on luxol fast stainings, thus we termed this region affected white matter.

Whether the ischemia-related cell types are typically represented in the ischemic core or in the peri-infarct area is indeed an important question. Therefore, we also quantified cell counts within the lesion core, for each reported staining, providing several additional supplementary figures, in our revised manuscript (see Supplementary Figs. 15,18,28,30,32).

The reviewer correctly pointed out that the MRI images in figures 3, 5 and 6 correspond to severe MCAO cases. Indeed, the vast of majority of MCAO cases included in the rat IF-cohort exhibit larger MCAO lesions than the moderate MCAO cohort used for snRNAseq (Supplementary Fig.1). Moreover, there is a wide SD in this cohort, unfortunately not uncommon for the filament model of permanent MCAO.

To provide a further validation of our findings, in confirmed perilesional areas in a separate and more moderate MCAO model, as suggested by the reviewer, we employed a different rodent model of MCAO. In the revised manuscript we introduced data derived from n=4 mice with Thrombin induced thromboembolic MCAO, 48h post lesion. As evidenced in Supplementary Fig.16, within this model significantly smaller lesions are induced, with less variation. Reassuringly, the vast majority of our findings was well conserved across species and MCAO models, providing a further experimental layer to strengthen the herein reported main findings (see Supplementary Figs. 17,19,29,31).

3) Although the English language and grammar is fine, the manuscript and the figures are a very heavy read. The figures are extremely busy, with several panels (often four) under each letter. The figures should be simplified by removing non-necessary panels (around 50% can be removed or transferred to supplement). The Results main text is also hard to follow. It should be simplified, and details can be moved to Methods or Supplement.

Response:

This point is well taken, we revised our manuscript in an effort to communicate our data to a broader audience and increase readability. Firstly, we avoided subpanels under individual letters within the figures, wherever reasonable, as requested. However, we refrained from excluding any subpanels because we propose that the information presented is necessary to

comprehensively describe the main findings of our study. For example, it might seem excessive to depict DEGs as Volcano Plots and also show many of these DEGs again within heatmaps, depicting their average expression and their occurrence in pseudo time analysis derived modules, within the same figure (for example Fig.2). However, these panels portray the results of different, complementary computational methods, thus allowing the reader to critically assess the presented data from several perspectives, at a glance. Importantly, the 7 main figures already are a condensation of 33 underlying supplementary figures and 11 Supplementary Data files, which include multiple datasheets.

Likewise, we understand that the used terminology in this manuscript might often sound overly technical. We attempted to reduce the use of specific technical jargon within the results as much as possible. However, we consciously decided to stay rather close to the terminology suggested by the authors of the herein used computational methods, specifically to minimize the risk of a misleading interpretation of our results. For example, the sentence “The aggregate rank score of module 9 increased with incremental distance to the trajectory bifurcation within MOLIGO_0, indicating a dynamic progression from the MOLIGO_0 towards the MOLIGO_1 cell state, specifically within infarcted tissue.” indeed is a heavy read. It might be rather tempting to rephrase this, for example to: “Pseudo time analysis indicated that within infarcted tissue a priori homeostatic mature Oligodendrocytes (MOLIGO_0) developed into the infarction specific MOLIGO_1 state”. Such phrasing however conceals to some degree that the progression from the MOLIGO_0 towards the MOLIGO_1 cell state is a computational inference. The actual result in this case is “the aggregate rank score of module 9 increased with incremental distance to the trajectory bifurcation within MOLIGO_0”, while “indicating a dynamic progression from the MOLIGO_0 towards the MOLIGO_1 cell state, specifically within infarcted tissue.” in this case describes what might be inferred from this result. Thus we propose that omitting the more technical description of the result, would often leave out the actual observed finding, jumping straight to the inferred conclusion.

This of course does not exempt us from the goal of communicating the main findings of our study comprehensibly. Thus we also added several descriptive paragraphs which summarize main findings, however taking care to not overly extend the result section excessively. Furthermore, we understand that it is frustrating for the reader to be forced to refer to multiple cited references in order to interpret the herein used computational methods and corresponding results. Therefore, we added several descriptive paragraphs to the method section, specifically to help readers, which are not routinely confronted with these methods to interpret our findings with less effort.

4) Fig 1D: The Olig2 group is much smaller in Sham R than in Sham L. I would have expected these to be similar. Needs to be discussed.

Response:

We share the reviewer's perplexity regarding this observation. We do not propose any biological difference between hemispheres regarding the differential abundance of OLIGO_2 cells, based on the observation that the nuclei of the OLIGO_2 cluster do not differ transcriptionally between the two hemispheres (0 DEGs were derived from comparing the transcriptome of OLIGO_2 between the Sham hemispheres, see Supplementary Fig.6a,b). It is thus most likely that this observation was caused by a sampling artefact, which we state in the revised version of our manuscript.

5) Cells were only sampled at one time point (48h). This is an important drawback and should be discussed more thoroughly.

Response:

Indeed, we fully agree with the reviewer, in that tracking the transcriptional profiles of reactive glial and immune cells across the maturation of the glial scar would be highly informative. In our revised manuscript we discuss this limitation more extensively. This limitation is now discussed on three separate occasions, at greater length in our revised manuscript.

6) The authors mention in the Discussion that some of the astrocyte results are "limited by the low number of captured AC_5 nuclei". However, it is not clear from the Results section that AC_nuclei was low. Does this mean AC_5 results are less reliable?

Response:

Nuclei numbers are depicted in main Fig.4c and Supplementary Fig.21c as stacked bar plots. The results section was amended to state more clearly that AC_3 was the largest among the reactive astrocyte clusters. The nuclei numbers in AC_4 and AC_5 are lower compared to AC_3, AC_3 also was more conserved across both stroke severities. Indeed, generally the reliability of a transcriptional phenotype increases with the number of sequenced cells/nuclei and the sequencing depth achieved for a respective (sub-)cluster. Thus AC_3 is the most robust reactive astrocyte cell state within our dataset. This does not mean, that smaller clusters are biologically irrelevant. Often smaller clusters constitute rare cell types or transient cell states. However, we decided to highlight the limitations arising from the smaller cluster size in some of the reactive astrocyte subclusters more diligently in our revised manuscript.

Reviewer #3 (Remarks to the Author):

Response

We are grateful for the reviewers invested time and effort.

Reviewer #4 (Remarks to the Author):

The paper by Daniel Bormann et al., titled "Single Nucleus RNA Sequencing Reveals Glial Cell Type-Specific Responses to Ischemic Stroke," explores the cellular landscape and microenvironment during the early stages of ischemic stroke recovery (48 hpi) in a rat stroke model. Utilizing snRNA-seq, the study uncovers molecular signatures in microglia, oligodendrocyte lineage cells, and astrocytes, elucidating their transcriptional perturbations during acute recovery. Notably, the authors characterize infarction-restricted proliferating oligodendrocyte precursor cells (OPCs), mature oligodendrocytes, and reactive astrocyte populations, revealing commonalities in their response to ischemic injury. The paper also explores immuno-glial cross talk with stroke-specific myeloid cells, particularly the involvement of osteopontin-positive myeloid cells and their interaction with CD44-positive OPCs and astrocytes. Finally, the in vitro experiment demonstrates the functional relevance of osteopontin in enhancing OPC migratory capacity.

The manuscript is well-written and provides a thorough analysis of the snRNA-seq data, offering valuable insights into cellular responses following ischemic stroke. The authors' approach in characterizing infarction-specific molecular signatures in oligodendrocyte lineage cells and astrocytes is commendable, documented meticulously in numerous figures and supplementary tables. The inclusion of immunohistochemistry to validate stroke-induced populations adds a robust experimental layer, reinforcing the reliability of the identified cell states. The exploration of cell-cell communication, particularly the shared immuno-glial cross talk between stroke-induced OPCs, astrocytes, and stroke-specific myeloid cells, is a noteworthy aspect of this study. The immunohistochemistry analysis of osteopontin-positive myeloid cells in close proximity to CD44-positive OPCs and reactive astrocytes within the perilesional zone, along with subsequent functional validation showing the enhancement of OPC migratory capacity after osteopontin treatment, adds a functional dimension to the findings.

In summary, I find no major objections to the manuscript and support its acceptance for publication. I have included some minor comments and suggestions for the authors' consideration, which I believe will further enhance the clarity and impact of the study.

Response:

We are grateful for the reviewers reassuring assessment of our manuscript, thorough review and insightful remarks. Below each suggestion is addressed point by point.

- The authors utilized a 5' sequencing approach, which is not standard in the field. Could they provide a rationale for this choice?

Response:

We are of course willing to elaborate on the question raised by the reviewer. The 5' sequencing approach offers the unique advantage of integration with other omics- applications, such as VDJ sequencing or CITE-seq. Because of this higher versatility several departments at our center jointly switched to the 5' sequencing approach kits. Although this aspect was not crucial for the herein presented dataset, the 10XGenomics 16 rxn kits at our center are often shared for different projects, with different demands.

Nevertheless, we carefully considered whether the switch from 3' to 5' sequencing might be disadvantageous for single nucleus sequencing of CNS tissue. Reassuringly, several recent high impact publications have successfully implemented the 5'kit, addressing single cell level transcriptional changes in the context of different neuropathologies^{8, 9, 10}

Moreover, we have previously used the 5'kit and thus far have not experienced any technical limitations when integrating previous datasets from our group generated by this approach with 3' independent sequencing datasets (see Direder et al. (2023)¹¹). Thus, we expect that researchers will not encounter any technical limitations when integrating our data with other scRNAseq or snRNAseq datasets in the future.

- While I concur with the authors' interpretation of the MC_OLIGO cluster, have they considered the possibility that these cells represent doublets?

Response:

The reviewer raises an important point, although we used the Double Finder pipeline and set upper UMI and Feature count cut offs to remove potential doublets before downstream analysis, some residual contaminating doublets might still remain in the final integrated dataset. Typically, doublets would be characterized by substantially higher UMI and Feature counts compared to the respective individual populations from which the doublet is composed, in this case oligodendrocyte lineage cells and myeloid cells. We have not observed dramatically increased UMI and Feature counts within the MC_OLIGO cluster, in fact the UMI and Feature Counts of the MC_OLIGO cluster most closely resemble the UMI and Feature count distribution of the activated MG_1 microglia cluster (see Reply Fig.2, below). This observation further corroborates on the hypothesis that this clusters harbors phagocytic myeloid cells, which accumulated oligodendrocyte transcripts in the nuclear compartment, as described previously¹². However, please note that we discuss further possible explanations, for the observed transcriptional profile of MC_OLIGO in the supplementary notes.

Reply Fig2. UMI and feature counts observed within MC_OLIGO resemble myeloid subcluster MG_1. a,b Number of unique transcripts (=UMIs), across oligodendrocyte lineage (a) and myeloid cell subclusters (b). **c,d** Gene (=feature) counts, across oligodendrocyte lineage (c) and myeloid cell subclusters (d). All metrics are reported for each sequenced sample, with y axes scales log normalized. Black bars denote median values, medians of MC_OLIGO and MG_1 highlighted with grey, dashed lines.

- Lines 649-650 state, "Pseudotime trajectory analysis indicated that the stroke-specific sub-cluster OPC_1 branched directly from the conserved sub-cluster OPC_0 (Fig.2b)." This claim may not be easily visible in the figure; clarification is suggested.

Response:

We thank the reviewer for this excellent suggestion. In the revised figure we have traced the pseudotime trajectory branches leading towards OPC_1, MOLIGO_1 and MOLIGO_2 in the respective color associated to this cluster and also labeled these clusters in the pseudotime feature plot to provide visual clues illustrating this statement in the results section.

- Have the authors explored the overlap of the OPC_1 cluster with any "disease-associated" cluster of OPC in the literature? Such a comparison could yield insights into similarities and differences across pathologies.

Response:

We concur with the reviewer's assessment. Pandey et al. (2022)¹³ from which we derived the disease-associated signatures have indeed described the emergence of DAO signatures in their study also in oligodendrocyte precursor cells (see Pandey et al. (2022) – "Figure S5: Emergence of disease-associated subtypes of oligodendrocyte precursor cells"). Note that the authors also observed the prominent upregulation of CD44 in these cells, similar to the stroke associated OPCs in our dataset. Curiously, the authors here also describe the emergence of proliferating "cycling OPCs", which however formed a separate cluster in their dataset. To our understanding the DAO like gene profile in OPCs in this study was not described as a separate specific DAO-OPC signature but rather an overlap to the general oligodendrocyte DAO signatures, which were included in our study.

Nevertheless, further transcriptional comparisons of the stroke associated OPC state in our dataset to other pathologies would indeed be possible and potentially highly interesting, for example regarding communalities and differences to traumatic brain injuries and spinal cord lesions. However, to address all other major concerns stated by the reviewers in an adequate manner we already extended the presented work to include 7 main figures and 33 Supplementary figures. Therefore, we reasoned that the inclusion of further transcriptional comparisons would be excessive and could overshadow our primary findings and decrease the overall readability of the paper. However, we strived to back each of our main results with detailed Supplementary Data and uploaded our raw and processed data to a publicly available repository, so that the inclined reader can easily follow up on such comparisons.

- Some descriptive paragraphs may benefit from concluding sentences to highlight major findings and improve overall readability

Response:

This point is well taken, we added several new concluding sentences to highlight major findings, as suggested.

- The authors might consider to highlight certain discussed ligand-receptors in Supplementary Figure 11, particularly the spp1-CD44 signaling pathway.

Response:

We agree, adding highlights might make it easier recognizing selected pathways mentioned in the main text. We therefore highlighted all pathways which contain Spp1 and/or CD44 with the respective figures, as requested (Supplementary Figs. 26 and 27, in the revised manuscript).

- A detailed analysis of human tissues, including quantification and comparison to unaffected tissue, could strengthen the paper. If revisited, this figure may warrant inclusion in the main text rather than the supplement.

Response:

Indeed, particularly data derived from early human infarctions, that are more closely modeled by our dataset, including various demographics would be desirable. Unfortunately, we currently are unable to obtain enough human samples for a more thorough investigation. Likewise, we currently lack appropriate region and age matched control tissue to revisit investigations in human tissue in more detail. Please note that the presented data from humans is rather heterogeneous in terms of affected anatomical localization and age groups. Therefore, we reasoned it to be more transparent to present these data as individual cases. However, we have provided novel data derived from a controlled experiment in which we obtained data from mice, 48h after induction of thromboembolic MCAO, to further strengthen the main results of our study.

- The suggestion to use the dash symbol "-" consistently for instances where single-cell or single-nucleus is used as an adjective is noted for improved consistency.

Response:

We thank the reviewer for the thorough review of our manuscript, we have edited our manuscript accordingly.

Reply References:

1. Sadick JS, O'Dea MR, Hasel P, Dykstra T, Faustin A, Liddelow SA. Astrocytes and oligodendrocytes undergo subtype-specific transcriptional changes in Alzheimer's disease. *Neuron* **110**, 1788-1805.e1710 (2022).
2. Beuker C, *et al.* Stroke induces disease-specific myeloid cells in the brain parenchyma and pia. *Nat Commun* **13**, 945 (2022).
3. Maitra M, *et al.* Extraction of nuclei from archived postmortem tissues for single-nucleus sequencing applications. *Nat Protoc* **16**, 2788-2801 (2021).
4. Nagy C, *et al.* Single-nucleus transcriptomics of the prefrontal cortex in major depressive disorder implicates oligodendrocyte precursor cells and excitatory neurons. *Nat Neurosci* **23**, 771-781 (2020).
5. Bormann D, *et al.* Exploring the heterogeneous transcriptional response of the CNS to systemic LPS and Poly(I:C). *Neurobiol Dis* **188**, 106339 (2023).
6. Wong B. Points of view: Color blindness. *Nature Methods* **8**, 441-441 (2011).
7. Bonfanti E, *et al.* The role of oligodendrocyte precursor cells expressing the GPR17 receptor in brain remodeling after stroke. *Cell Death Dis* **8**, e2871 (2017).
8. Hou J, *et al.* Transcriptomic atlas and interaction networks of brain cells in mouse CNS demyelination and remyelination. *Cell Rep* **42**, 112293 (2023).
9. Zhou Y, *et al.* Human early-onset dementia caused by DAP12 deficiency reveals a unique signature of dysregulated microglia. *Nat Immunol* **24**, 545-557 (2023).
10. Zhou Y, *et al.* Human and mouse single-nucleus transcriptomics reveal TREM2-dependent and TREM2-independent cellular responses in Alzheimer's disease. *Nat Med* **26**, 131-142 (2020).
11. Direder M, *et al.* The transcriptional profile of keloidal Schwann cells. *Exp Mol Med* **54**, 1886-1900 (2022).
12. Schirmer L, *et al.* Neuronal vulnerability and multilineage diversity in multiple sclerosis. *Nature* **573**, 75-82 (2019).
13. Pandey S, *et al.* Disease-associated oligodendrocyte responses across neurodegenerative diseases. *Cell Rep* **40**, 111189 (2022).

Reviewers' Comments:

Reviewer #1:

Remarks to the Author:

The authors have sufficiently addressed all of comments, I have no additional concerns arising from the revision

Reviewer #2:

Remarks to the Author:

The authors have answered all my concerns and have improved the manuscript accordingly. I have no more comments and I believe the manuscript is ready for publication.

Reviewer #3:

Remarks to the Author:

I co-reviewed this manuscript with one of the reviewers who provided the listed reports. This is part of the Nature Communications initiative to facilitate training in peer review and to provide appropriate recognition for Early Career Researchers who co-review manuscripts

Reviewer #4:

Remarks to the Author:

The authors have addressed all my comments, and I am happy to recommend the manuscript for publication.

REVIEWERS' COMMENTS

Reviewer #1 (Remarks to the Author):

The authors have sufficiently addressed all of comments, I have no additional concerns arising from the revision

Reviewer #2 (Remarks to the Author):

The authors have answered all my concerns and have improved the manuscript accordingly. I have no more comments and I believe the manuscript is ready for publication.

Reviewer #3 (Remarks to the Author):

I co-reviewed this manuscript with one of the reviewers who provided the listed reports. This is part of the Nature Communications initiative to facilitate training in peer review and to provide appropriate recognition for Early Career Researchers who co-review manuscripts

Reviewer #4 (Remarks to the Author):

The authors have addressed all my comments, and I am happy to recommend the manuscript for publication.

Response: We thank the reviewers for their constructive feedback during this review process and are delighted that the revised manuscript has met the reviewers expectations.